# A Unified Discretization Framework for Differential Equation Approach with Lyapunov Arguments for Convex Optimization

**Kansei Ushiyama**
The University of Tokyo, Tokyo, Japan
ushiyama-kansei074@g.ecc.u-tokyo.ac.jp

**Shun Sato**
The University of Tokyo, Tokyo, Japan
shun@mist.i.u-tokyo.ac.jp

**Takayasu Matsuo**
The University of Tokyo, Tokyo, Japan
matsuo@mist.i.u-tokyo.ac.jp

## Abstract

The differential equation (DE) approach for convex optimization, which relates optimization methods to specific continuous DEs with rate-revealing Lyapunov functionals, has gained increasing interest since the seminal paper by Su–Boyd–Candès (2014). However, the approach still lacks a crucial component to make it truly useful: there is no general, consistent way to transition back to discrete optimization methods. Consequently, even if we derive insights from continuous DEs, we still need to perform individualized and tedious calculations for the analysis of each method. This paper aims to bridge this gap by introducing a new concept called "weak discrete gradient" (wDG), which consolidates the conditions required for discrete versions of gradients in the DE approach arguments. We then define abstract optimization methods using wDG and provide abstract convergence theories that parallel those in continuous DEs. We demonstrate that many typical optimization methods and their convergence rates can be derived as special cases of this abstract theory. The proposed unified discretization framework for the differential equation approach to convex optimization provides an easy environment for developing new optimization methods and achieving competitive convergence rates with state-of-the-art methods, such as Nesterov's accelerated gradient.

## 1 Introduction

In this paper, we consider unconstrained convex optimization problems:

$$\min_{x \in \mathbb{R}^d} f(x). \tag{1}$$

Various optimization methods, such as standard gradient descent and Nesterov's accelerated gradient methods (Nesterov, 1983), are known for these problems. The convergence rates of these methods have been intensively investigated based on the classes of objective functions ($L$-smooth and/or $\mu$-strongly convex). We focus on the convergence rate of function values $f\big(x^{(k)}\big) - f^\star$, while the rates for $\|\nabla f\big(x^{(k)}\big)\|$ or $\|x^{(k)} - x^\star\|$ have also been discussed. Topics particularly relevant to this study include the lower bound of convergence rates for first-order methods (see Remark 4.3 for the relationship between our framework and first-order methods) for convex and strongly convex functions: $\mathrm{O}\big(1/k^2\big)$ for $L$-smooth and convex functions (cf. Nesterov (2018)) and $\mathrm{O}\Big(\big(1 - \sqrt{\mu/L}\big)^{2k}\Big)$ for $L$-smooth and $\mu$-strongly convex functions (Drori and Taylor, 2022). These lower bounds are tight, as they are

37th Conference on Neural Information Processing Systems (NeurIPS 2023).

achieved by some optimization methods, such as Nesterov (1983) for convex functions and Van Scoy et al. (2018); Taylor and Drori (2022a) for strongly convex functions. In these studies, the discussion is typically conducted for each method, utilizing various techniques accumulated in the optimization research field.

Whereas, it has long been known that some optimization methods can be related to continuous differential equations (DEs). Early works on this aspect include the following: the continuous gradient flow $\dot{x} = -\nabla f(x)$ as a continuous optimization method was discussed in Bruck (1975). Similar arguments were later applied to second-order differential equations (Alvarez, 2000; Alvarez et al., 2002; Cabot et al., 2009). An important milestone in this direction was Su et al. (2014), where it was shown that Nesterov's famous accelerated gradient method (Nesterov, 1983) could be related to a second-order system with a convergence rate-revealing "Lyapunov functional." The insights gained from this relationship have been useful in understanding the behavior of the Nesterov method and in considering its new variants. This success has followed by many studies, including Wilson (2018); Wilson et al. (2021). The advantage of the DEs with Lyapunov functional approach (which we simply call the "DE approach" hereafter) is that the continuous DEs are generally more intuitive, and convergence rate estimates are quite straightforward thanks to the Lyapunov functionals. However, the DE approach still lacks one important component; although we can draw useful insights from continuous DEs, there is no known general way to translate them into a discrete setting. Consequently, we still need to perform complex discrete arguments for each method. This limitation was already acknowledged in Su et al. (2014): "... The translation, however, involves parameter tuning and tedious calculations. This is the reason why a general theory mapping properties of ODEs into corresponding properties for discrete updates would be a welcome advance."

In this paper we attempt to provide this missing piece by incorporating the concept of "discrete gradients" (DGs) from numerical analysis, which is used to replicate some properties of continuous DEs in discrete settings. We demonstrate that a relaxed concept of DG, which we call "weak discrete gradient" (wDG), can serve a similar purpose in the optimization context. More precisely, we show that for known DEs in the DE approach, if we define abstract optimization methods using wDGs analogously to the DEs, their abstract convergence theories can be obtained by following the continuous arguments and replacing gradients with wDGs. The tedious parts of the case-specific discrete arguments are consolidated in the definition of wDG, which simplifies the overall arguments: *we can now consider "simple continuous DE arguments" and "case-specific discrete discussions summarized in wDG" separately.* We demonstrate that many typical existing optimization methods and their rate estimates, previously done separately for each method, can be recovered as special cases of the abstract methods/theories, providing a simpler view of them. Any untested combination of a known DE and wDG presents an obvious new method and its rate, further expanding the potential for innovation in the optimization field. Creating a new wDG leads to a series of optimization methods by applying it to known DEs. One simply needs to verify if the wDG satisfies the conditions for wDG (Theorem 4.2) and reveal the constants of the wDG. If, in the future, a new DE with a rate-revealing Lyapunov functional is discovered, it should be possible to achieve similar results. We suggest first defining an abstract wDG method analogous to the DE and then examining whether the continuous theory can be translated to a discrete setting, as demonstrated in this paper.

The aforementioned paper (Su et al., 2014) concludes in the following way (continued from the previous quote) "Indeed, this would allow researchers to only study the simpler and more user-friendly ODEs." Although there is still room for minor adjustments (see the discussion on limitations below), we believe the wDG framework substantially reduces the complexity of discussions in discrete methods, allowing researchers to focus on more accessible and intuitive aspects of optimization.

We consider the problems (1) on the $d$-dimensional Euclidean space $\mathbb{R}^d$ ($d$ is a positive integer) with the standard inner product $\langle \cdot, \cdot \rangle$ and the induced norm $\|\cdot\|$, where $f \colon \mathbb{R}^d \to \mathbb{R}$ represents a differentiable convex objective function. We assume the existence of the optimal value $f^\star$ and the optimal solution $x^\star$. In the following discussion, we use the inequality

$$\frac{\mu}{2}\|y - x\|^2 \leq f(y) - f(x) - \langle \nabla f(x), y - x \rangle, \tag{2}$$

which holds for any $x, y \in \mathbb{R}^d$ when $f$ is $\mu$-strongly convex and differentiable.

*Remark* 1.1. Although the scope of this paper is limitted due to the restriction of space, the framework can be naturally extended to more general cases. Extension to the objective functions satisfying the Polyak–Łojasiewicz (PŁ) condition is provided in Appendix H. The framework can be extended to

constrained optimizations by the DE approach for mirror descent methods (cf. Krichene et al. (2015); Wilson et al. (2021)), which the authors have already confirmed. Stochastic methods such as the stochastic gradient descent can be handled by considering random compositions of wDGs, which is left as our future work.

## 2   Short summary of the differential equation approach

Let us first consider the gradient flow:

$$\dot{x} = -\nabla f(x), \quad x(0) = x_0 \in \mathbb{R}^d. \tag{3}$$

It is easy to see that

$$\frac{\mathrm{d}}{\mathrm{d}t} f(x(t)) = \langle \nabla f(x(t)), \dot{x}(t) \rangle = -\|\nabla f(x(t))\|^2 \leq 0. \tag{4}$$

This means the flow can be regarded as a continuous optimization method. Notice that the proof is quite simple, once we admit the *chain rule of differentiation*, and the *form of the flow* itself (3); this will be quite important in the subsequent discussion.

Despite its simplicity, the convergence rate varies depending on the class of objective functions. Below we show some known results. The following rates are proven using the so-called Lyapunov argument, which introduces a "Lyapunov functional" that explicitly contains the convergence rate. The proof is left to Appendix B (we only note here that, in addition to the two key tools the chain rule and the form of the flow we need the *convexity inequality* (2) to complete the proof.)

**Theorem 2.1** (Convex case). *Suppose that $f$ is convex. Let $x\colon [0,\infty) \to \mathbb{R}^d$ be the solution of the gradient flow* (3). *Then the solution satisfies*

$$f(x(t)) - f^\star \leq \frac{\|x_0 - x^\star\|^2}{2t}.$$

**Theorem 2.2** (Strongly convex case). *Suppose that $f$ is $\mu$-strongly convex. Let $x\colon [0,\infty) \to \mathbb{R}^d$ be the solution of the gradient flow* (3). *Then the solution satisfies*

$$f(x(t)) - f^\star \leq \mathrm{e}^{-\mu t}\|x_0 - x^\star\|^2.$$

An incomplete partial list of works using the Lyapunov approach includes, in addition to Su et al. (2014), Karimi and Vavasis (2016); Attouch et al. (2016); Attouch and Cabot (2017); Attouch et al. (2018); França et al. (2018); Defazio (2019); Shi et al. (2019); Wilson et al. (2021) (see also a comprehensive list in Suh et al. (2022)). A difficulty in this approach is that the Lyapunov functionals were found only heuristically. A remedy is provided in Suh et al. (2022); Du (2022), but its target is still limited.

Next, we consider DEs corresponding to accelerated gradient methods, including Nesterov's method. As is well known, the forms of accelerated gradient methods differ depending on the class of objective functions, and consequently, the DEs to be considered also change. In this paper, we call them *accelerated gradient flows*.

When the objective functions are convex, we consider the following DE proposed in Wilson et al. (2021): let $A\colon \mathbb{R}_{\geq 0} \to \mathbb{R}_{\geq 0}$ be a differentiable strictly monotonically increasing function with $A(0) = 0$, and

$$\dot{x} = \frac{\dot{A}}{A}(v - x), \qquad \dot{v} = -\frac{\dot{A}}{4}\nabla f(x), \tag{5}$$

with $(x(0), v(0)) = (x_0, v_0) \in \mathbb{R}^d \times \mathbb{R}^d$.

**Theorem 2.3** (Convex case (Wilson et al. (2021))). *Suppose that $f$ is convex. Let $(x, v)\colon [0,\infty) \to \mathbb{R}^d \times \mathbb{R}^d$ be the solution of the DE* (5). *Then it satisfies*

$$f(x(t)) - f^\star \leq \frac{2\|x_0 - x^\star\|^2}{A(t)}.$$

*Remark* 2.4. If we set $A(t) = t^2$, this system coincides with a continuous limit DE of the accelerated gradient method for convex functions

$$\ddot{x} + \frac{3}{t}\dot{x} + \nabla f(x) = 0,$$

which is derived in Su et al. (2016).

*Remark* 2.5. From Theorem 2.3, it might seem that we can achieve an arbitrarily high order rate. Although it is surely true in the continuous context, it does not imply we can construct discrete optimization methods from the ODE. In fact, greedily demanding a higher rate is penalized at the timing of discretization from the numerical stability. See, for example, the discussion in Ushiyama et al. (2022b).

Next, for strongly convex objective functions, let us consider the DE (again in Wilson et al. (2021)):

$$\dot{x} = \sqrt{\mu}(v - x), \qquad \dot{v} = \sqrt{\mu}(x - v - \nabla f(x)/\mu) \tag{6}$$

with $(x(0), v(0)) = (x_0, v_0) \in \mathbb{R}^d \times \mathbb{R}^d$. (Note that this system coincides with the continuous limit ODE of the accelerated gradient method for strongly convex functions by Polyak (1964): $\ddot{x} + 2\sqrt{\mu}\dot{x} + \nabla f(x) = 0$.)

**Theorem 2.6** (Strongly convex case (Wilson et al. (2021); Luo and Chen (2022))). *Suppose that $f$ is $\mu$-strongly convex. Let $(x, v)\colon [0, \infty) \to \mathbb{R}^d \times \mathbb{R}^d$ be the solution of* (6). *Then it satisfies*

$$f(x(t)) - f^\star \leq e^{-\sqrt{\mu}t}\left(f(x_0) - f^\star + \frac{\mu}{2}\|v_0 - x^\star\|^2\right).$$

## 3 Discrete gradient method for gradient flows (from numerical analysis)

The remaining issue is how we discretize the above DEs. In the optimization context, it was done separately in each study. One tempting strategy for a more systematic discretization is to import the concept of "DG," which was invented in numerical analysis for designing structure-preserving numerical methods for gradient flows such as (3) (Gonzalez (1996); McLachlan et al. (1999)). Recall that the automatic decrease of objective function came from the two keys: (a) the chain rule, and (b) the gradient flow structure. The DG method respects and tries to imitate them in discrete settings.

**Definition 3.1** (Discrete gradient (Gonzalez (1996); Quispel and Capel (1996))). A continuous map $\nabla_{\mathrm{d}}f\colon \mathbb{R}^d \times \mathbb{R}^d \to \mathbb{R}^d$ is said to be *discrete gradient of $f$* if the following two conditions hold for all $x, y \in \mathbb{R}^d$:

$$f(y) - f(x) = \langle \nabla_{\mathrm{d}}f(y, x), y - x\rangle, \qquad \nabla_{\mathrm{d}}f(x, x) = \nabla f(x). \tag{7}$$

In the definition provided above, the second condition simply requires that $\nabla_{\mathrm{d}}f$ approximates $\nabla f$. On the contrary, the first condition, referred to as the *discrete chain rule*, is a critical requirement for the key (a). The discrete chain rule is a scalar equality constraint on the vector-valued function, and for any given $f$, there are generally infinitely many DGs. The following is a list of some popular choices of DGs. When it is necessary to differentiate them from wDGs, we call them *strict DGs*.

**Proposition 3.2** (Strict discrete gradients). *The following functions are strict DGs.*
*Gonzalez discrete gradient $\nabla_{\mathrm{G}}f(y, x)$ (Gonzalez (1996)):*

$$\nabla f\left(\frac{y+x}{2}\right) + \frac{f(y) - f(x) - \langle\nabla f\left(\frac{y+x}{2}\right), y - x\rangle}{\|y-x\|^2}(y - x).$$

*Itoh–Abe discrete gradient $\nabla_{\mathrm{IA}}f(y, x)$ (Itoh and Abe (1988)):*

$$\begin{bmatrix} \frac{f(y_1, x_2, x_3\ldots, x_d) - f(x_1, x_2, x_3, \ldots, x_d)}{y_1 - x_1} \\ \frac{f(y_1, y_2, x_3\ldots, x_d) - f(y_1, x_2, x_3, \ldots, x_d)}{y_2 - x_2} \\ \vdots \\ \frac{f(y_1, y_2, y_3, \ldots, y_d) - f(y_1, y_2, y_3\ldots, x_d)}{y_d - x_d} \end{bmatrix}.$$

*Average vector field (AVF) $\nabla_{\mathrm{AVF}}f(y, x)$ (Quispel and McLaren (2008)):*

$$\int_0^1 \nabla f(\tau y + (1 - \tau)x)\mathrm{d}\tau.$$

Suppose we have a DG for a given $f$. Then we can define a discrete scheme for the gradient flow (3):

$$\frac{x^{(k+1)} - x^{(k)}}{h} = -\nabla_{\mathrm{d}} f\left(x^{(k+1)}, x^{(k)}\right), \quad x^{(0)} = x_0,$$

where the positive real number $h$ is referred to as the step size, and $x^{(k)} \simeq x(kh)$ is the numerical solution. The left-hand side approximates $\dot{x}$ and is denoted by $\delta^+ x^{(k)}$ hereafter. Note that the definition conforms to the key point (b) mentioned earlier.

The scheme decreases $f(x^{(k)})$ as expected:

$$\left(f\left(x^{(k+1)}\right) - f\left(x^{(k)}\right)\right) / h = \left\langle \nabla_{\mathrm{d}} f\left(x^{(k+1)}, x^{(k)}\right), \delta^+ x^{(k)}\right\rangle = -\left\|\nabla_{\mathrm{d}} f\left(x^{(k+1)}, x^{(k)}\right)\right\|^2 \leq 0.$$

In the first equality we used the discrete chain rule, and in the second, the form of the scheme itself. Observe that the proof proceeds in the same manner as the continuous case (4). Due to the decreasing property, the scheme should work as an optimization method. Additionally, the above argument does not reply on the step size $h$, and it can be changed in every step (which will not destroy the decreasing property).

In the numerical analysis community, the above approach has already been attempted for optimizations (Grimm et al. (2017); Celledoni et al. (2018); Ehrhardt et al. (2018); Miyatake et al. (2018); Ringholm et al. (2018); Benning et al. (2020); Riis et al. (2022)). Although they were successful on their own, this does not immediately provide the missing piece we seek for the following reasons. First, the DG framework does not include typical important optimization methods; it even does not include the steepest descent. Second, as noted above, the proofs of rate estimates in the continuous DEs (in Section 2) require the inequality of convexity (2). Unfortunately, however, existing DGs generally do not satisfy it; see Appendix C for a counterexample. Next, we show how to overcome these difficulties.

*Remark* 3.3. Some members of the optimization community may find the use of DGs peculiar, since it involves referring to two solutions $x^{(k+1)}, x^{(k)}$. However, in some sense, it is quite natural because the decrease of $f$ occurs in a single step $x^{(k)} \mapsto x^{(k+1)}$. There may also be concerns about the computational complexity of DGs because the method becomes "implicit" by referring to $x^{(k+1)}$. In the field of structure-preserving numerical methods, however, it is widely known that in some highly unstable DEs, implicit methods are often advantageous, allowing larger time-stepping widths, while explicit methods require extremely small ones. In fact, it has been confirmed in Ehrhardt et al. (2018) that this also applies to the optimization context. The Itoh–Abe DG results in a system of $d$ nonlinear equations, which is slightly less expensive. Moreover, note that the integral in the AVF can be evaluated analytically before implementation, when $f$ is a polynomial.

## 4   Weak discrete gradients and abstract optimization methods

We introduce the concept of *a weak discrete gradient* (wDG), which is a relaxed version of the DG introduced earlier.

**Definition 4.1** (Weak discrete gradient). A gradient approximation[1] $\overline{\nabla} f \colon \mathbb{R}^d \times \mathbb{R}^d \to \mathbb{R}^d$ is said to be *weak discrete gradient of $f$* if there exists $\alpha \geq 0$ and $\beta, \gamma$ with $\beta + \gamma \geq 0$ such that the following two conditions hold for all $x, y, z \in \mathbb{R}^d$:

$$f(y) - f(x) \leq \left\langle \overline{\nabla} f(y, z), y - x\right\rangle + \alpha\|y - z\|^2 - \beta\|z - x\|^2 - \gamma\|y - x\|^2, \quad \overline{\nabla} f(x, x) = \nabla f(x). \tag{8}$$

Note that (8) can be regarded as a modification of the three points descent lemma, where the third variable z is utilized to give some estimates. The freedom in variable z in (8) is fully utilized also in this paper; see Theorems 5.4 and 5.5 and their proofs.

The condition (8) can be interpreted in two ways. First, it can be understood as a discrete chain rule in a weaker sense. By substituting $x$ with $z$, we obtain the inequality:

$$f(y) - f(x) \leq \left\langle \overline{\nabla} f(y, x), y - x\right\rangle + (\alpha - \gamma)\|y - x\|^2. \tag{9}$$

---

[1] Notice that we use the notation $\overline{\nabla}$ here, distinguishing it from $\nabla_{\mathrm{d}}$ denoted as the standard notation for strict discrete gradients in numerical analysis.

Compared to the strict discrete chain rule (7), it is weaker because it is an inequality and allows an error term. Second, it can be interpreted as a weaker discrete convex inequality. By exchanging $x$ and $y$ and rearranging terms, we obtain another expression

$$f(y) - f(x) - \left\langle \overline{\nabla} f(x,z), y - x \right\rangle \geq \gamma\|y - x\|^2 + \beta\|y - z\|^2 - \alpha\|x - z\|^2. \qquad (10)$$

Compared to the strongly convex inequality (2), the term $(\mu/2)\|y - x\|^2$ is now replaced with $\beta\|y - x\|^2 + \gamma\|y - z\|^2$, which can be interpreted as the squared distance between $y$ and the point $(x,z)$ where the gradient is evaluated. The term $-\alpha\|x - z\|^2$ is an error term.

We now list some examples of wDGs (proof is provided in Appendix D). Notice that these examples include various typical gradient approximations from the optimization and numerical analysis literature. Note that for ease of presentation, we simply write "$\mu$-strongly convex function," which includes convex functions by setting $\mu = 0$.

**Theorem 4.2.** *Suppose that $f\colon \mathbb{R}^d \to \mathbb{R}$ is a $\mu$-strongly convex function. Let* (L) *and* (SC) *denote the additional assumptions:* (L) *$f$ is $L$-smooth, and* (SC) *$\mu > 0$. Then, the following functions are wDGs:*

(i) *If $\overline{\nabla} f(y,x) = \nabla f(x)$ and $f$ satisfies* (L)*, then $(\alpha, \beta, \gamma) = (L/2, \mu/2, 0)$.*

(ii) *If $\overline{\nabla} f(y,x) = \nabla f(y)$, then $(\alpha, \beta, \gamma) = (0, 0, \mu/2)$.*

(iii) *If $\overline{\nabla} f(y,x) = \nabla f(\frac{x+y}{2})$ and $f$ satisfies* (L)*, then $(\alpha, \beta, \gamma) = ((L+\mu)/8, \mu/4, \mu/4)$.*

(iv) *If $\overline{\nabla} f(y,x) = \nabla_{\mathrm{AVF}} f(y,x)$ and $f$ satisfies* (L)*, then $(\alpha, \beta, \gamma) = (L/6 + \mu/12, \mu/4, \mu/4)$.*

(v) *If $\overline{\nabla} f(y,x) = \nabla_{\mathrm{G}} f(y,x)$ and $f$ satisfies* (L)(SC)*, then $(\alpha, \beta, \gamma) = ((L+\mu)/8 + (L-\mu)^2/16\mu, \mu/4, 0)$.*

(vi) *If $\overline{\nabla} f(y,x) = \nabla_{\mathrm{IA}} f(y,x)$ and $f$ satisfies* (L)(SC)*, then $(\alpha, \beta, \gamma) = (dL^2/\mu - \mu/4, \mu/2, -\mu/4)$.*

Although we assumed the smoothness of $f$ to simplify the presentation, the case (ii) does not demand it (see the end of Appendix D). Thus, it can handle non-smooth convex optimization. While the wDGs (i), (iii), (iv) only require (L), the wDGs (v) and (vi) demand (SC) ($\mu > 0$). This implies that the latter wDGs might be fragile for small $\mu$'s.

We now define an abstract method using wDGs:

$$\frac{x^{(k+1)} - x^{(k)}}{h} = -\overline{\nabla} f\left(x^{(k+1)}, x^{(k)}\right), \quad x^{(0)} = x_0, \qquad (11)$$

which is analogous to the gradient flow (3). By "abstract," we mean that it is a formal formula, and given a concrete wDG it reduces to a concrete method; see Table 1 which summarizes some typical choices. Observe that the abstract method covers many popular methods from both optimization and numerical analysis communities. The step size $h$ may be selected using line search techniques, but for simplicity, we limit our presentation to the fixed step size in this paper (see Remark 5.1).

*Remark* 4.3. Note that some wDGs are not directly connected to the original gradient $\nabla f$'s; the Itoh–Abe wDG (vi) does not even refer to the gradient. Thus, the concrete methods resulting from our framework do not necessarily fall into the so-called "first-order methods," which run in a linear space spanned by the past gradients (Nesterov (1983)). This is why we use the terminology "gradient-based methods" in this paper, instead of first-order methods.

Similar to the aforementioned, we can define abstract methods for (5) and (6). Details and theoretical results can be found in Theorems 5.4 and 5.5.

We also introduce the next lemma, which is useful in expanding the scope of our framework.

**Lemma 4.4.** *Suppose $f$ can be expressed as a sum of two functions $f_1, f_2$. If $\overline{\nabla}_1 f_1$ and $\overline{\nabla}_2 f_2$ are wDGs of $f_1$ and $f_2$ with parameters $(\alpha_1, \beta_1, \gamma_1)$ and $(\alpha_2, \beta_2, \gamma_2)$, respectively, then $\overline{\nabla}_1 f_1 + \overline{\nabla}_2 f_2$ is a weak discrete gradient of $f$ with $(\alpha, \beta, \gamma) = (\alpha_1 + \alpha_2, \beta_1 + \beta_2, \gamma_1 + \gamma_2)$.*

This lemma allows us to consider the following discretization of the gradient flow:

$$\frac{x^{(k+1)} - x^{(k)}}{h} = -\nabla f_1\left(x^{(k)}\right) - \nabla f_2\left(x^{(k+1)}\right) \qquad (12)$$

within our framework. For instance, if $f_1$ is $L_1$-smooth and $\mu_1$-strongly convex, and $f_2$ is $\mu_2$-strongly convex, then the right-hand side of (12) is a wDG with $(\alpha, \beta, \gamma) = (L_1/2, \mu_1/2, \mu_2/2)$. In this case, the method is known as the proximal gradient method or the forward-backward splitting algorithm in optimization (cf. Bauschke and Combettes (2017)). Discretizing the accelerated gradient flows allows for obtaining accelerated versions. (Acceleration of the proximal gradient method has been studied for some time (Beck and Teboulle, 2009b,a).)

Table 1: Examples of wDGs and their corresponding convergence rates for a $\mu$-strongly convex and $L$-smooth function $f$ on $\mathbb{R}^d$. The numbers in the $\overline{\nabla} f$ column correspond to the numbers in Theorem 4.2. The line in the figure corresponding to the proximal gradient method is described in the setting of (12). The notation (DG) represents a strict discrete gradient. The convergence rates shown in the table are the best possible for the step sizes chosen in Theorems 5.3 and 5.5.

| | Opt. meth. | | Convergence rates | |
| --- | --- | --- | --- | --- |
| $\overline{\nabla} f$ | (for (3)) | Numer. meth. | Theorem 5.3 | Theorem 5.5 |
| (i) | steep. des. | exp. Euler | $O\left(\left(1 - 2\frac{\mu}{L+\mu}\right)^k\right)$ | $O\left(\left(1 - \sqrt{\frac{\mu}{L}}\right)^k\right)$ |
| (ii) | prox. point | imp. Euler | $0$ | $0$ |
| (i)+(ii) | prox. grad. | (splitting) | $O\left(\left(1 - 2\frac{\mu_1+\mu_2}{L_1+\mu_1+2\mu_2}\right)^k\right)$ | $O\left(\left(1 - \sqrt{\frac{\mu_1+\mu_2}{L_1+\mu_2}}\right)^k\right)$ |
| (iii) | — | imp. midpoint | $O\left(\left(1 - 8\frac{\mu}{L+7\mu}\right)^k\right)$ | $O\left(\left(1 - \sqrt{\frac{4\mu}{L+3\mu}}\right)^k\right)$ |
| (iv) | — | AVF (DG) | $O\left(\left(1 - 6\frac{\mu}{L+5\mu}\right)^k\right)$ | $O\left(\left(1 - \sqrt{\frac{3\mu}{L+2\mu}}\right)^k\right)$ |
| (v) | — | Gonzalez (DG) | $O\left(\left(1 - 8\frac{\mu^2}{L^2+4\mu^2}\right)^k\right)$ | $O\left(\left(1 - \sqrt{\frac{4\mu^2}{L^2+3\mu^2}}\right)^k\right)$ |
| (vi) | — | Itoh–Abe (DG) | $O\left(\left(1 - 2\frac{\mu^2}{4d^2L^2-\mu^2}\right)^k\right)$ | $O\left(\left(1 - \sqrt{\frac{\mu^2}{4dL^2-2\mu^2}}\right)^k\right)$ |

## 5 Convergence rates of abstract optimization methods

We establish the discrete counterparts of Theorems 2.1 to 2.3 and 2.6. Although the proofs are left to Appendix E, we emphasize that they can be performed analogously to those of the continuous cases. The discrete theorems are established in four cases: the gradient flow (3) (for $f$ convex and $\mu$-strongly convex), and the accelerated flows (for $f$ convex (5) and $\mu$-strongly convex (6)). For ease of understanding, we summarize the results for $\mu$-strongly convex cases in Table 1. A similar table for convex cases is included in Appendix A.

*Remark* 5.1. The following theorems are presented under the assumption that the step size $h$ is fixed for simplicity. However, if all varying step sizes satisfy the step size condition (with a finite number of violations allowed), the theorems still hold true. The step sizes must be bounded by a positive number from below to ensure the designated rates.)

### 5.1 For the abstract method based on the gradient flow

The abstract method is given in (11).

**Theorem 5.2** (Convex case). *Let $f$ be a convex function. Let $\overline{\nabla} f$ be a wDG of $f$, and suppose that $f$ also satisfies the necessary conditions required by the wDG. Suppose that in the wDG $\beta \geq 0, \gamma \geq 0$. Let $\{x^{(k)}\}$ be the sequence given by* (11). *Then, under the step size condition $h \leq 1/(2\alpha)$, the sequence satisfies*

$$f\left(x^{(k)}\right) - f^\star \leq \frac{\|x_0 - x^\star\|^2}{2kh}.$$

Let us demonstrate how to use the theorem using the proximal gradient method (12) as an example. Suppose that $f_1$ is $L_1$-smooth and convex, and $f_2$ is convex. Then, $\overline{\nabla} f(y, x) = \nabla f_1(x) + \nabla f_2(y)$ is a wDG with the parameter $(\alpha, \beta, \gamma) = (L_1/2, 0, 0)$ due to Theorem 4.2 and Lemma 4.4. Therefore,

the proximal gradient method (12) satisfies the assumption of Theorem 5.2 and thus the convergence rate is $\mathrm{O}(1/k)$ under the step size condition $h \leq (1/L_1)$.

**Theorem 5.3** (Strongly convex case)**.** *Let $f$ be a strongly convex function. Let $\overline{\nabla} f$ be a wDG of $f$, and suppose that $f$ also satisfies the necessary conditions required by the wDG. Suppose that in the wDG $\beta + \gamma > 0$. Let $\{x^{(k)}\}$ be the sequence given by* (11)*. Then, under the step size condition $h \leq 1/(\alpha + \beta)$, the sequence satisfies*

$$f\left(x^{(k)}\right) - f^\star \leq \left(1 - \frac{2(\beta + \gamma)h}{1 + 2\gamma h}\right)^k \|x_0 - x^\star\|^2.$$

*In particular, the sequence satisfies*

$$f\left(x^{(k)}\right) - f^\star \leq \left(1 - \frac{2(\beta + \gamma)}{\alpha + \beta + 2\gamma}\right)^k \|x_0 - x^\star\|^2,$$

*when the optimal step size $h = 1/(\alpha + \beta)$ is employed.*

## 5.2 For the abstract methods based on the accelerated gradient flows

We consider abstract methods with wDGs based on the accelerated gradient flows (5) and (6), which will be embedded in the theorems below. We note one thing: when using (8) as an approximation of the chain rule, we can determine $z$ independently of $x$ and $y$, which gives us some degrees of freedom (thus allowing for adjustment.) Below we show some choices of $z^{(k)}$ that are easy to calculate from known values while keeping the decrease of the Lyapunov functional.

**Theorem 5.4** (Convex case)**.** *Let $f$ be a convex function. Let $\overline{\nabla} f$ be a wDG of $f$, and suppose that $f$ also satisfies the necessary conditions required by the wDG. Suppose that in the wDG $\beta \geq 0, \gamma \geq 0$. Let $\left\{\left(x^{(k)}, v^{(k)}\right)\right\}$ be the sequence given by*

$$\begin{cases} \delta^+ x^{(k)} = \dfrac{\delta^+ A_k}{A_k}\left(v^{(k+1)} - x^{(k+1)}\right), \\[2mm] \delta^+ v^{(k)} = -\dfrac{\delta^+ A_k}{4}\overline{\nabla} f\left(x^{(k+1)}, z^{(k)}\right), \\[2mm] \dfrac{z^{(k)} - x^{(k)}}{h} = \dfrac{\delta^+ A_k}{A_{k+1}}\left(v^{(k)} - x^{(k)}\right) \end{cases}$$

*with $\left(x^{(0)}, v^{(0)}\right) = (x_0, v_0)$, where $A_k := A(kh)$. Then if $A_k = (kh)^2$ and $h \leq 1/\sqrt{2\alpha}$, the sequence satisfies*

$$f\left(x^{(k)}\right) - f^\star \leq \frac{2\|x_0 - x^\star\|^2}{A_k}.$$

**Theorem 5.5** (Strongly convex case)**.** *Let $f$ be a strongly convex function. Let $\overline{\nabla} f$ be a wDG of $f$, and suppose that $f$ also satisfies the necessary conditions required by the wDG. Suppose that in the wDG $\beta + \gamma > 0$. Let $\left\{\left(x^{(k)}, v^{(k)}\right)\right\}$ be the sequence given by*

$$\begin{cases} \delta^+ x^{(k)} = \sqrt{2(\beta + \gamma)}\left(v^{(k+1)} - x^{(k+1)}\right), \\[2mm] \delta^+ v^{(k)} = \sqrt{2(\beta + \gamma)}\left(\dfrac{\beta}{\beta + \gamma}z^{(k)} + \dfrac{\gamma}{\beta + \gamma}x^{(k+1)} - v^{(k+1)} - \dfrac{\overline{\nabla} f\left(x^{(k+1)}, z^{(k)}\right)}{2(\beta + \gamma)}\right), \\[2mm] \dfrac{z^{(k)} - x^{(k)}}{h} = \sqrt{2(\beta + \gamma)}\left(x^{(k)} + v^{(k)} - 2z^{(k)}\right) \end{cases}$$

*with $\left(x^{(0)}, v^{(0)}\right) = (x_0, v_0)$. Then if $h \leq \overline{h} := \left(\sqrt{2}(\sqrt{\alpha + \gamma} - \sqrt{\beta + \gamma})\right)^{-1}$, the sequence satisfies*

$$f\left(x^{(k)}\right) - f^\star \leq \left(1 + \sqrt{2(\beta + \gamma)}h\right)^{-k}\left(f(x_0) - f^\star + \beta\|v_0 - x^\star\|^2\right).$$

*In particular, the sequence satisfies*

$$f\left(x^{(k)}\right) - f^\star \leq \left(1 - \sqrt{\frac{\beta + \gamma}{\alpha + \gamma}}\right)^k\left(f(x_0) - f^\star + \beta\|v_0 - x^\star\|^2\right),$$

*when the optimal step size $h = \overline{h}$ is employed.*

*Remark* 5.6. Time scaling can eliminate the factor $\sqrt{2(\beta + \gamma)}$ from the scheme and simplify it, as shown in Luo and Chen (2022). However, we do not use time scaling here to match the time scale with the accelerated gradient method and to maintain correspondence with the continuous system.

# 6   Discussions including Limitations

**Relation to some other systematic/unified frameworks**   A systematic approach to obtaining optimization methods with convergence estimates was developed using the "performance estimation problems" (PEPs) technique, as seen in works such as Taylor et al. (2018); Taylor and Drori (2022b). While our framework unifies discussions in both continuous and discrete settings, the design of methods is not automatic and requires finding a new wDG. In contrast, the PEP framework automates method design but separates discussions between the continuous and discrete settings. Combining these two approaches could be a promising research direction, such as applying our framework to the Lyapunov functionals obtained in Taylor and Drori (2022b).

Another unified convergence analysis is presented in Chen and Luo (2021), where the authors cite the same passage in the Introduction from Su et al. (2016). However, this seems to focus on unifying discussions in the continuous setting, and it is still necessary to individualize the discretization for each method in the discrete setting.

In Diakonikolas and Orecchia (2019), a unified method for deriving continuous DEs describing first-order optimization methods was proposed using the "approximate duality gap technique." While this work is capable of finding new DEs, it does not provide insight into how to discretize the DEs for obtaining discrete optimization methods.

Another closely related work is De Sa et al. (2022), which proposed a framework to construct Lyapunov functionals for continuous ODEs. This is strong in view of the fact that generally Lyapunov functionals can be found only in ad hoc ways. Instead, they considered only the simplest gradient descent (and its stochastic version), while the main focus of the present paper lies in the discretizations.

As said in Section 3, in the field of numerical analysis, the use of discrete gradients has been tried. Among them, Ehrhardt et al. (2018) is a pioneering work that comes with several theoretical results. Both this and the present work aim at convex, strongly convex, and the PŁ functions (in the Appendix, in the present paper). The scope of Ehrhardt et al. (2018) was limited in the sense that they considered only discretizations of gradient flows with strict discrete gradients. Our target ODEs and discretizations are not limited to that, but as its price, our rate is worse in some strict discrete gradient discretizations of gradient flow. This comes from the difference in proof techniques: they proved convergence rates directly and algebraically, while our analysis is via Lyapunov functionals. They also gave several theoretical results besides the convergence analysis, such as the (unique) existence of solutions, and step-size analysis which are important in actual implementations. Whether these two frameworks could be unified would be an interesting future research topic.

**Limitations of the proposed framework.**   Although we believe that the current framework provides an easy environment for working on the DE approach to optimization, it still has some limitations.

First, methods that do not fall into the current framework exist, such as the following splitting method (cf. the Douglas–Rachford splitting method (Eckstein and Bertsekas, 1992)):

$$\frac{x^{(k+1/2)} - x^{(k)}}{h} = -\overline{\nabla}_1 f_1\Big(x^{(k+1/2)}, x^{(k)}\Big), \quad \frac{x^{(k+1)} - x^{(k+1/2)}}{h} = -\overline{\nabla}_2 f_2\Big(x^{(k+1)}, x^{(k+1/2)}\Big).$$

The right-hand side cannot be written by a single wDG. Additionally, methods based on Runge–Kutta (RK) numerical methods (Zhang et al. (2018); Ushiyama et al. (2022a)) appear difficult to be captured by wDG because RK methods cannot be expressed by DG in the first place. Investigating whether these methods can be captured by the concept of DG is an interesting future research topic.

Second, there is still some room for adjustment in wDG methods. A typical example is $z^{(k)}$ in Section 5.2, which is chosen in the theorems to optimize efficiency and rates. Another example is the adjustment of time-stepping in the last phase of constructing a method to achieve a better rate or practical efficiency. Although these optimizations in the construction of optimization methods are standard in optimization studies, we feel that they are difficult to capture in the current framework, as they fall between the intuitive continuous argument and the discrete wDG arguments that aim to capture common structures.

Third, some rates in the theorems are not optimal. For example, on strongly convex functions, the scheme proposed in Theorem 5.5 with the choice (i) achieves the convergence rate of $\mathrm{O}\left((1 - \sqrt{\mu/L})^k\right)$, which is not the optimal rate of $\mathrm{O}\left((1 - \sqrt{\mu/L})^{2k}\right)$. This is because the choice of the DE and Lyapunov functional used in this work is not optimal. A DE and Lyapunov functional for obtaining the optimal rate are known (Sun et al., 2020), but the DE is a so-called high-resolution DE (known as a "modified equation" in numerical analysis), which involves the Hessian. Whether these DEs can be captured with the wDG perspective is an interesting future research topic.

## 7 Concluding remaks

In this paper, we proposed a new unified discretization framework for the DE approach to convex optimization. Our framework provides an easy environment for those working on the DE approach, and some new methods are immediate from the framework, both as methods and for their convergence estimates. For example, any combination of strict DGs with the accelerated gradient flows are new methods, and their rates are given by the theorems. Although we did not include numerical experiments in the main body owing to the space restrictions, some preliminary numerical tests confirming the theory can be found in Appendix I. These tests show that some new methods can be competitive with state-of-the-art methods, such as Nesterov's accelerated gradient.

## Acknowledgements

The authors are grateful for the anonymous reviewers who gave helpful comments to improve this paper. This work is partially supported by JSPS KAKENHI Grant Number JP22K13955, JP21H03452, JP20K21786, and JP20H01822, and by JST SPRING Grant Number JPMJSP2108.

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

# A  Table summarizing the convex cases

In contrast to the strongly convex cases, in the convex cases the convergence rates are the same in the order, and the difference appears in its coefficients, which actually depends on the maximum step sizes allowed. See Theorems 5.2 and 5.4 for the rates. In the table below we summarize the step size information.

Table 2: Examples of weak discrete gradients and resulting convergence rates when $f$ is an $L$-smooth function on $\mathbb{R}^d$. The numbers in the $\overline{\nabla} f$ column correspond to the numbers in Theorem 4.2. The line of the proximal gradient method is described in the setting of (12). (DG) represents that this weak discrete gradient is a strict discrete gradient.

| | Opt. meth. | | Max. step sizes | |
| $\overline{\nabla} f$ | (for (3)) | Numer. meth. | Theorem 5.3 | Theorem 5.5 |
|---|---|---|---|---|
| (i) | steep. des. | exp. Euler | $1/L$ | $1/\sqrt{L}$ |
| (ii) | prox. point | imp. Euler | $\infty$ | $\infty$ |
| (i)+(ii) | prox. grad. | (splitting) | $1/L_1$ | $1/\sqrt{L_1}$ |
| (iii) | — | imp. midpoint | $4/L$ | $2/\sqrt{L}$ |
| (iv) | — | AVF (DG) | $3/L$ | $\sqrt{3/L}$ |

# B  Proofs of theorems in Section 2

## B.1  Proof of Theorem 2.1

It is sufficient to show that a Lyapunov function

$$E(t) := t(f(x(t)) - f^\star) + \frac{1}{2}\|x(t) - x^\star\|^2$$

is monotonically nonincreasing since

$$f(x(t)) - f^\star \leq \frac{E(t)}{t} \leq \frac{E(0)}{t} = \frac{\|x(0) - x^\star\|^2}{2t}.$$

Indeed,

$$\begin{aligned}
\dot{E} &= t\langle \nabla f(x), \dot{x}\rangle + f(x) - f^\star + \langle x - x^\star, \dot{x}\rangle \\
&= -t\|\nabla f(x)\|^2 + f(x) - f^\star - \langle \nabla f(x), x - x^\star\rangle \\
&\leq 0
\end{aligned}$$

holds. Here, at each line, we applied the Leibniz rule and the chain rule, substituted the ODE, and used the convexity of $f$ in this order. ∎

## B.2  Proof of Theorem 2.2

It is sufficient to show a Lyapunov function

$$E(t) := e^{\mu t}\left(f(x(t)) - f^\star + \frac{\mu}{2}\|x(t) - x^\star\|^2\right)$$

is monotonically nonincreasing and thus it is sufficient to show that $\tilde{E}(t) := e^{-\mu t}E(t)$ satisfies $\dot{\tilde{E}} \leq -\mu\tilde{E}$. Indeed,

$$\begin{aligned}
\dot{\tilde{E}} &= \langle \nabla f(x), \dot{x}\rangle + \mu\langle x - x^\star, \dot{x}\rangle \\
&= -\|\nabla f(x)\|^2 - \mu\langle \nabla f(x), x - x^\star\rangle \\
&\leq -\|\nabla f(x)\|^2 - \mu\left(f(x) - f^\star + \frac{\mu}{2}\|x - x^\star\|^2\right) \\
&\leq -\mu\tilde{E}
\end{aligned}$$

holds. Here, at each line, we applied the chain rule, substituted the ODE, and used strong convexity of $f$ in this order. ∎

### B.3 Proof of Theorem 2.3

It is sufficient to show that

$$E(t) := A(t)(f(x(t)) - f^\star) + 2\|v(t) - x^\star\|^2$$

is nonincreasing. Actually,

$$
\begin{aligned}
\dot{E} &= \dot{A}(f(x) - f^\star) + A\frac{\mathrm{d}}{\mathrm{d}t}(f(x) - f^\star) + \frac{\mathrm{d}}{\mathrm{d}t}(2\|v(t) - x^\star\|^2) \\
&= \dot{A}(f(x) - f^\star) + A\langle \nabla f(x), \dot{x}\rangle + 4\langle \dot{v}, v - x^\star\rangle \\
&= \dot{A}(f(x) - f^\star) + A\left\langle \nabla f(x), \frac{\dot{A}}{A}(v - x)\right\rangle - 4\left\langle \frac{\dot{A}}{4}\nabla f(x), v - x^\star\right\rangle \\
&= \dot{A}(f(x) - f^\star - \langle \nabla f(x), x - x^\star\rangle) \\
&\le 0
\end{aligned}
$$

holds. Here, at each line, we applied the Leibniz rule, used the chain rule, substituted the ODE, and used the convexity of $f$ in this order. ∎

### B.4 Proof of Theorem 2.6

It is sufficient to show that

$$E(t) := \mathrm{e}^{\sqrt{\mu}t}\left(f(x) - f^\star + \frac{\mu}{2}\|v - x^\star\|^2\right)$$

is nonincreasing and thus it is sufficient to show that $\tilde{E}(t) := \mathrm{e}^{-\sqrt{\mu}t}E(t)$ satisfies $\dot{\tilde{E}} \le -\sqrt{\mu}\tilde{E}$. Actually,

$$
\begin{aligned}
\dot{\tilde{E}} &= \langle \nabla f(x), \dot{x}\rangle + \mu\langle \dot{v}, v - x^\star\rangle \\
&= \langle \nabla f(x), \sqrt{\mu}(v - x)\rangle + \mu\langle \sqrt{\mu}(x - v - \nabla f(x)/\mu), v - x^\star\rangle \\
&= \sqrt{\mu}(\langle \nabla f(x), x^\star - x\rangle - \mu\langle v - x, v - x^\star\rangle) \\
&= \sqrt{\mu}\left(\langle \nabla f(x), x^\star - x\rangle - \frac{\mu}{2}(\|v - x\|^2 + \|v - x^\star\|^2 - \|x - x^\star\|^2)\right) \\
&\le -\sqrt{\mu}\left(\left(f(x) - f^\star + \frac{\mu}{2}\|v - x^\star\|^2\right) - \frac{\mu}{2}\|v - x\|^2\right) \\
&\le -\sqrt{\mu}\tilde{E}
\end{aligned}
$$

holds. Here, at each line, we applied the chain rule, substituted the ODE, rearranged terms, decomposed the inner product by the law of cosines (see Appendix F), and used the strong convexity of $f$. ∎

## C  Strict discrete gradients and convexity

In this section, we describe that strict discrete gradients are not generally compatible with the convex inequality; this complements the discussion in Section 3. For example, let us consider imitating the discussion in Appendix B.1 by using a discrete gradient scheme $\delta^+x^{(k)} = -\nabla_\mathrm{d}f\big(x^{(k+1)}, x^{(k)}\big)$. Then, since we use the inequality

$$f(x) - f^\star - \langle \nabla f(x), x - x^\star\rangle \le 0$$

that holds due to the convexity of $f$, we should ensure that the discrete counterpart of the left-hand side

$$f(x) - f^\star - \langle \nabla_\mathrm{d}f(y, x), x - x^\star\rangle$$

is nonpositive for any $x, y \in \mathbb{R}^d$. However, there is a simple counterexample as shown below.

Let us consider a quadratic and convex objective function $f(x) = \frac{1}{2}\langle x, Qx\rangle$, where $Q \in \mathbb{R}^{d\times d}$ is a positive definite matrix. In this case, $x^\star = 0$ and $f^\star = 0$ hold. Then, when we choose a discrete gradient $\nabla_\mathrm{d}f(y, x) = \nabla_\mathrm{G}f(y, x) = \nabla_\mathrm{AVF}f(y, x) = Q\left(\frac{y+x}{2}\right)$, we see

$$f(x) - f^\star - \langle \nabla_\mathrm{d}f(y, x), x - x^\star\rangle = \frac{1}{2}\langle x, Qx\rangle - \left\langle Q\left(\frac{y+x}{2}\right), x\right\rangle = -\frac{1}{2}\langle y, Qx\rangle,$$

which is positive when $y = -x$ and $x \ne 0$.

## D  Proof of Theorem 4.2

(i) Since we assume $f$ is $L$-smooth and $\mu$-strongly convex,

$$f(y) - f(z) \le \langle \nabla f(z), y - z \rangle + \frac{L}{2}\|y - z\|^2,$$

$$f(z) - f(x) \le \langle \nabla f(z), z - x \rangle - \frac{\mu}{2}\|z - x\|^2$$

holds for any $x, y, z \in \mathbb{R}^d$. By adding each side of these inequalities, we obtain

$$f(y) - f(x) \le \langle \nabla f(z), y - x \rangle + \frac{L}{2}\|y - z\|^2 - \frac{\mu}{2}\|z - x\|^2. \tag{13}$$

(This inequality is known as the three points descent lemma in optimization.)

(ii) It follows immediately from the $\mu$-strong convexity of $f$.

(iii) By replacing $z$ in (13) with $\theta y + (1 - \theta)z$, and invoking Lemma F.2, we have

$$f(y) - f(x) - \langle \nabla f(\theta y + (1-\theta)z), y - x \rangle$$

$$\le \frac{L}{2}\|y - (\theta y + (1-\theta)z)\|^2 - \frac{\mu}{2}\|\theta y + (1-\theta)z - x\|^2$$

$$= \frac{L}{2}(1-\theta)^2\|y - z\|^2 - \frac{\mu}{2}(-\theta(1-\theta)\|y - z\|^2 + (1-\theta)\|z - x\|^2 + \theta\|y - x\|^2).$$

Especially when $\theta = 1/2$, $(\alpha, \beta, \gamma) = (L/8 + \mu/8, \mu/4, \mu/4)$.

(iv) By the same calculation as (iii), we obtain

$$f(y) - f(x) - \left\langle \int_0^1 \nabla f(\tau y + (1-\tau)z)\mathrm{d}\tau, y - x \right\rangle$$

$$= \int_0^1 \left[ f(y) - f(x) - \langle \nabla f(\tau y + (1-\tau)z), y - x \rangle \right] \mathrm{d}\tau$$

$$\le \int_0^1 \left[ \frac{L}{2}(1-\tau)^2\|y - z\|^2 - \frac{\mu}{2}(-\tau(1-\tau)\|y - z\|^2 + (1-\tau)\|z - x\|^2 + \tau\|y - x\|^2) \right] \mathrm{d}\tau$$

$$= \left( \frac{L}{6} + \frac{\mu}{12} \right)\|y - z\|^2 - \frac{\mu}{4}\|z - x\|^2 - \frac{\mu}{4}\|y - x\|^2.$$

(v) By (13), we obtain

$$f(y) - f(z) - \left\langle \nabla f\left(\frac{y + z}{2}\right), y - z \right\rangle \le \frac{L}{2}\left\|y - \frac{y+z}{2}\right\|^2 - \frac{\mu}{2}\left\|\frac{y+z}{2} - z\right\|^2 = \frac{L - \mu}{8}\|y - z\|^2.$$

Since this inequality holds with $y$ and $z$ swapped, we have

$$\left| f(y) - f(z) - \left\langle \nabla f\left(\frac{y + z}{2}\right), y - z \right\rangle \right| \le \frac{L - \mu}{8}\|y - z\|^2.$$

Thus,

$$f(y) - f(x) - \langle \nabla_{\mathrm{G}} f(y, z), y - x \rangle$$

$$= f(y) - f(x) - \left\langle \nabla f\left(\frac{y + z}{2}\right) + \frac{f(y) - f(z) - \langle \nabla f\left(\frac{y+z}{2}\right), y - z \rangle}{\|y - z\|^2}(y - z), y - x \right\rangle$$

$$\le f(y) - f(x) - \left\langle \nabla f\left(\frac{y + z}{2}\right), y - x \right\rangle + \left| \frac{f(y) - f(z) - \langle \nabla f\left(\frac{y+z}{2}\right), y - z \rangle}{\|y - z\|^2} \right| |\langle y - z, y - x \rangle|$$

$$\le \frac{L}{2}\left\|y - \frac{y + z}{2}\right\|^2 - \frac{\mu}{2}\left\|\frac{y + z}{2} - x\right\|^2 + \frac{L - \mu}{8}\left( \frac{L - \mu}{8\mu}\|y - z\|^2 + \frac{2\mu}{L - \mu}\|y - x\|^2 \right)$$

$$= \frac{L}{8}\|y - z\|^2 - \frac{\mu}{2}\left( \frac{1}{2}\|y - x\|^2 + \frac{1}{2}\|z - x\|^2 - \frac{1}{4}\|y - z\|^2 \right) + \frac{(L - \mu)^2}{16\mu}\|y - z\|^2 + \frac{\mu}{4}\|y - x\|^2$$

$$= \left( \frac{L}{8} + \frac{\mu}{8} + \frac{(L - \mu)^2}{16\mu} \right)\|y - z\|^2 - \frac{\mu}{4}\|z - x\|^2$$

holds, where the second inequality follows from the arithmetic-geometric means (AM-GM) inequality.

(vi) In the following, for $y, z \in \mathbb{R}^d$ and $k = 2, \ldots, d$, $y_{1:k-1} z_{k:d}$ denotes a vector $(y_1, \ldots, y_{k-1}, z_k, \ldots, z_d)^\top \in \mathbb{R}^d$, while $y_{1:0} z_{1:d}$ and $y_{1:d} z_{d+1:d}$ denote $z$ and $y$, respectively. By the telescoping sum and $\mu$-strong convexity of $f$, we obtain

$$f(y) - f(x) - \langle \nabla_{\mathrm{IA}} f(y, z), y - x \rangle$$

$$= f(y) - f(x) - \sum_{k=1}^d \frac{f(y_{1:k} z_{k+1:d}) - f(y_{1:k-1} z_{k:d})}{y_k - z_k}(y_k - x_k)$$

$$= f(y) - f(x) - \sum_{k=1}^d \frac{f(y_{1:k} z_{k+1:d}) - f(y_{1:k-1} z_{k:d})}{y_k - z_k}(y_k - z_k + z_k - x_k)$$

$$= f(z) - f(x) - \sum_{k=1}^d \frac{f(y_{1:k} z_{k+1:d}) - f(y_{1:k-1} z_{k:d})}{y_k - z_k}(z_k - x_k)$$

$$\leq \langle \nabla f(z), z - x \rangle - \frac{\mu}{2}\|z - x\|^2$$

$$+ \sum_{k=1}^d \begin{cases} \dfrac{(\nabla f(y_{1:k-1} z_{k:d}))_k(z_k - y_k) - \frac{\mu}{2}(z_k - y_k)^2}{y_k - z_k}(z_k - x_k) & \text{if } \dfrac{z_k - x_k}{y_k - z_k} > 0 \\ \dfrac{(\nabla f(y_{1:k} z_{k+1:d}))_k(y_k - z_k) - \frac{\mu}{2}(y_k - z_k)^2}{z_k - y_k}(z_k - x_k) & \text{if } \dfrac{z_k - x_k}{y_k - z_k} < 0 \end{cases}$$

$$= \langle \nabla f(z), z - x \rangle - \frac{\mu}{2}\|z - x\|^2$$

$$- \sum_{k=1}^d \begin{cases} (\nabla f(y_{1:k-1} z_{k:d}))_k(z_k - x_k) + \frac{\mu}{2}(y_k - z_k)(z_k - x_k) & \text{if } \dfrac{z_k - x_k}{y_k - z_k} > 0 \\ (\nabla f(y_{1:k} z_{k+1:d}))_k(z_k - x_k) + \frac{\mu}{2}(z_k - y_k)(z_k - x_k) & \text{if } \dfrac{z_k - x_k}{y_k - z_k} < 0 \end{cases}$$

$$= \sum_{k=1}^d \begin{cases} ((\nabla f(z))_k - (\nabla f(y_{1:k-1} z_{k:d}))_k)(z_k - x_k) & \text{if } \dfrac{z_k - x_k}{y_k - z_k} > 0 \\ ((\nabla f(z))_k - (\nabla f(y_{1:k} z_{k+1:d}))_k)(z_k - x_k) & \text{if } \dfrac{z_k - x_k}{y_k - z_k} < 0 \end{cases}$$

$$- \frac{\mu}{2}\|z - x\|^2 - \frac{\mu}{2}\sum_{k=1}^d |z_k - y_k||z_k - x_k|.$$

To evaluate the first term of the most right-hand side, we use the following inequalities:

$$|(\nabla f(z))_k - (\nabla f(y_{1:k-1} z_{k:d}))_k| \leq L\|z - y_{1:k-1} z_{k:d}\| \leq L\|z - y\|,$$
$$|(\nabla f(z))_k - (\nabla f(y_{1:k} z_{k+1:d}))_k| \leq L\|z - y_{1:k} z_{k+1:d}\| \leq L\|z - y\|,$$

which hold due to the $L$-smoothness of $f$, and also

$$\sum_{k=1}^d |z_k - x_k| \leq \sqrt{d \sum_{k=1}^d |z_k - x_k|^2} = \sqrt{d}\|z - x\|,$$

which holds due to Jensen's inequality. Using them, we obtain

$$f(y) - f(x) - \langle \overline{\nabla}_{\mathrm{IA}} f(y, z), y - x \rangle$$

$$\leq \sqrt{d}L\|z - y\|\|z - x\| - \frac{\mu}{2}\|z - x\|^2 - \frac{\mu}{2}\sum_{k=1}^d |z_k - y_k||z_k - x_k|$$

$$\leq \frac{dL^2}{\mu}\|z - y\|^2 + \frac{\mu}{4}\|z - x\|^2 - \frac{\mu}{2}\|z - x\|^2 - \frac{\mu}{2}\sum_{k=1}^d |z_k - y_k||z_k - x_k|,$$

where the last inequality holds due to the AM-GM inequality. Finally, the last term is bounded by

$$\frac{\mu}{2}\sum_{k=1}^{d}|z_k - y_k||z_k - x_k| = -\frac{\mu}{4}\sum_{k=1}^{d}\left(|z_k - y_k|^2 + |z_k - x_k|^2 - ||z_k - y_k| - |z_k - x_k||^2\right)$$

$$\leq -\frac{\mu}{4}\sum_{k=1}^{d}\left(|z_k - y_k|^2 + |z_k - x_k|^2 - |y_k - x_k|^2\right)$$

$$= -\frac{\mu}{4}\left(\|z - y\|^2 + \|z - x\|^2 - \|y - x\|^2\right).$$

This proves the theorem.

∎

For (ii), as noted in the above proof, the assumption of differentiability of $f$ is unnecessary, let alone $L$-smoothness. Since $f$ is a proper convex function on $\mathbb{R}^d$ in our setting, the subdifferential $\partial f(x)$ is nonempty for all $x \in \mathbb{R}^d$. Thus we can use $\overline{\nabla}f(y,x) \in \partial f(y)$ instead of $\overline{\nabla}f(y,x) = \nabla f(y)$. By definition of subgradients, we can recover the same parameters $(\alpha, \beta, \gamma)$ as the differentiable case.

If $\mu = 0$, the proofs for (v) and (vi) cease to work where we apply the AM-GM inequality to the inner product. This is also pointed out in the main body of the paper.

## E  Proofs of theorems in Section 5

### E.1  Proof of Theorem 5.2

It is sufficient to show that the discrete Lyapunov function

$$E^{(k)} := kh\left(f\left(x^{(k)}\right) - f^\star\right) + \frac{1}{2}\left\|x^{(k)} - x^\star\right\|^2$$

is nonincreasing. Actually,

$\delta^+ E^{(k)}$

$$= kh\left(\delta^+ f\left(x^{(k)}\right)\right) + f\left(x^{(k+1)}\right) - f^\star + \delta^+\left(\frac{1}{2}\left\|x^{(k)} - x^\star\right\|^2\right)$$

$$\leq kh\left(\left\langle\overline{\nabla}f\left(x^{(k+1)}, x^{(k)}\right), \delta^+ x^{(k)}\right\rangle + \alpha h\left\|\delta^+ x^{(k)}\right\|^2\right) + f\left(x^{(k+1)}\right) - f^\star + \left\langle x^{(k+1)} - x^\star, \delta^+ x^{(k)}\right\rangle - \frac{h}{2}\left\|\delta^+ x^{(k)}\right\|^2$$

$$= -kh(1 - \alpha h)\left\|\overline{\nabla}f\left(x^{(k+1)}, x^{(k)}\right)\right\|^2 + f\left(x^{(k+1)}\right) - f^\star - \left\langle\overline{\nabla}f\left(x^{(k+1)}, x^{(k)}\right), x^{(k+1)} - x^\star\right\rangle - \frac{h}{2}\left\|\delta^+ x^{(k)}\right\|^2$$

$$\leq -kh(1 - \alpha h)\left\|\overline{\nabla}f\left(x^{(k+1)}, x^{(k)}\right)\right\|^2 - \left(\frac{h}{2} - \alpha h^2\right)\left\|\delta^+ x^{(k)}\right\|^2$$

holds, and thus if $h \leq 1/(2\alpha)$, the right-hand side is not positive. Here, at each line, we applied the discrete Leibniz rule, the weak discrete gradient condition (8), Lemma F.1 as the chain rule, substituted the scheme, and applied again (8) as the convex inequality. ∎

### E.2  Proof of Theorem 5.3

Let

$$\tilde{E}^{(k)} := f\left(x^{(k)}\right) - f^\star + (\beta + \gamma)\left\|x^{(k)} - x^\star\right\|^2.$$

If $\delta^+ \tilde{E}^{(k)} \leq -c\tilde{E}^{(k+1)}$ for $c > 0$, it can be concluded that $E^{(k)} = (1 + ch)^k \tilde{E}^{(k)}$ is nonincreasing, and hence $f(x^{(k)}) - f^\star \leq (1 + ch)^{-k} E^{(0)}$. Actually,

$\delta^+ \tilde{E}^{(k)}$

$$
\begin{aligned}
&= \delta^+ f\left(x^{(k)}\right) + \delta^+ \left((\beta + \gamma)\left\|x^{(k)} - x^\star\right\|^2\right) \\
&\leq \left\langle \overline{\nabla} f\left(x^{(k+1)}, x^{(k)}\right), \delta^+ x^{(k)} \right\rangle + (\alpha - \gamma)h\left\|\delta^+ x^{(k)}\right\|^2 + 2(\beta + \gamma)\left\langle x^{(k+1)} - x^\star, \delta^+ x^{(k)} \right\rangle - (\beta + \gamma)h\left\|\delta^+ x^{(k)}\right\|^2 \\
&= -(1 - (\alpha - \gamma)h + (\beta + \gamma)h)\left\|\overline{\nabla} f\left(x^{(k+1)}, x^{(k)}\right)\right\|^2 - 2(\beta + \gamma)\left\langle x^{(k+1)} - x^\star, \overline{\nabla} f\left(x^{(k+1)}, x^{(k)}\right) \right\rangle \\
&\leq -2(\beta + \gamma)\left(f\left(x^{(k)}\right) - f^\star + \beta\left\|x^{(k)} - x^\star\right\|^2 + \gamma\left\|x^{(k+1)} - x^\star\right\|^2\right) \\
&\quad - \left(1 - (\alpha - \gamma)h + (\beta + \gamma)h - 2\alpha(\beta + \gamma)h^2\right)\left\|\overline{\nabla} f\left(x^{(k+1)}, x^{(k)}\right)\right\|^2 \quad\quad (14)
\end{aligned}
$$

holds. Here, after the second line we used the weak discrete gradient condition (8) as the chain rule, substituted the scheme and used (8) as the strongly convex inequality.

Now we aim to bound $\left\|x^{(k)} - x^\star\right\|^2$ with $\left\|x^{(k+1)} - x^\star\right\|^2$. By the same calculation for $\delta^+ \left\|x^{(k)} - x^\star\right\|^2$ as above, we get the evaluation

$$
\delta^+ \left\|x^{(k)} - x^\star\right\|^2 \leq -2(f(x^{(k+1)}) - f^\star + \beta\left\|x^{(k)} - x^\star\right\|^2 + \gamma\left\|x^{(k+1)} - x^{(k)}\right\|^2) - (h - 2\alpha h^2)\left\|\overline{\nabla} f\left(x^{(k+1)}, x^{(k)}\right)\right\|^2.
$$
$$(15)$$

Thus, if $h \leq 1/(2\alpha)$, we get $\left\|x^{(k+1)} - x^\star\right\|^2 \leq \left\|x^{(k)} - x^\star\right\|^2$. In this case, since the second term of (14) is nonpositive, it follows that

$$
\delta^+ \tilde{E}^{(k)} \leq -2(\beta + \gamma)\tilde{E}^{(k+1)}.
$$

To obtain a better rate which is included in the statement of the theorem, by directly using (15) for (14), we see

$$
\begin{aligned}
\delta^+ \tilde{E}^{(k)} \leq &-\frac{2(\beta + \gamma)}{1 - 2\beta h}\left(f(x^{(k+1)}) - f^\star + (\beta + \gamma)\left\|x^{(k+1)} - x^\star\right\|^2\right) \\
&- \left(\frac{1 - 2\alpha h}{1 - 2\beta h}2(\beta + \gamma)\beta h^2 + 1 - (\alpha - \gamma)h + (\beta + \gamma)h - 2\alpha(\beta + \gamma)h^2\right)\left\|\overline{\nabla} f\left(x^{(k+1)}, x^{(k)}\right)\right\|^2.
\end{aligned}
$$

Since the second term of the right-hand side is nonpositive under $h \leq 1/(\alpha + \beta)$, it can be concluded that

$$
\delta^+ \tilde{E}^{(k)} \leq -\frac{2(\beta + \gamma)}{1 - 2\beta h}\tilde{E}^{(k+1)}.
$$

In this case the convergence rate is

$$
\left(\frac{1}{1 + \frac{2(\beta + \gamma)h}{1 - 2\beta h}}\right)^k = \left(1 - \frac{2(\beta + \gamma)h}{1 + 2\gamma h}\right)^k.
$$

■

### E.3 Proof of Theorem 5.4

It is sufficient to show that

$$
E^{(k)} := A_k\left(f\left(x^{(k)}\right) - f^\star\right) + 2\|v^{(k)} - x^\star\|^2
$$

is nonincreasing. Actually,

$\delta^+ E^{(k)}$

$$= (\delta^+ A_k)\left(f\left(x^{(k+1)}\right) - f^\star\right) + A_k\left(\delta^+ f\left(x^{(k)}\right)\right) + 2\delta^+\left(\left\|v^{(k)} - x^\star\right\|^2\right)$$

$$\leq (\delta^+ A_k)\left(f\left(x^{(k+1)}\right) - f^\star\right) + A_k\left\langle \overline{\nabla} f\left(x^{(k+1)}, z^{(k)}\right), \delta^+ x^{(k)}\right\rangle$$

$$\quad + 4\left\langle \delta^+ v^{(k)}, v^{(k+1)} - x^\star\right\rangle + \frac{A_k}{h}\alpha\left\|x^{(k+1)} - z^{(k)}\right\|^2 - 2h\left\|\delta^+ v^{(k)}\right\|^2$$

$$\leq (\delta^+ A_k)\left(f\left(x^{(k+1)}\right) - f^\star\right) + A_k\left\langle \overline{\nabla} f\left(x^{(k+1)}, z^{(k)}\right), \frac{\delta^+ A_k}{A_k}\left(v^{(k+1)} - x^{(k+1)}\right)\right\rangle$$

$$\quad - 4\left\langle \frac{\delta^+ A_k}{4}\overline{\nabla} f\left(x^{(k+1)}, z^{(k)}\right), v^{(k+1)} - x^\star\right\rangle + \frac{A_k}{h}\alpha\left\|x^{(k+1)} - z^{(k)}\right\|^2 - 2h\left\|\delta^+ v^{(k)}\right\|^2$$

$$= (\delta^+ A_k)\left(f\left(x^{(k+1)}\right) - f^\star - \left\langle \overline{\nabla} f\left(x^{(k+1)}, z^{(k)}\right), x^{(k+1)} - x^\star\right\rangle\right) + \frac{A_k}{h}\alpha\left\|x^{(k+1)} - z^{(k)}\right\|^2 - 2h\left\|\delta^+ v^{(k)}\right\|^2$$

$$\leq (\delta^+ A_k)\alpha\left\|x^{(k+1)} - z^{(k)}\right\|^2 + \frac{A_k}{h}\alpha\left\|x^{(k+1)} - z^{(k)}\right\|^2 - 2h\left\|\delta^+ v^{(k)}\right\|^2$$

$$= \frac{1}{h}\left(A_{k+1}\alpha\left\|x^{(k+1)} - z^{(k)}\right\|^2 - 2\left\|v^{(k+1)} - v^{(k)}\right\|^2\right) =: \text{(err)}.$$

Here, at each line, we used the discrete Leibniz rule, applied (8) and Lemma F.1 as the chain rule, substituted the scheme, and applied again (8) as the convex inequality.

Now we define $z^{(k)}$ so that (err) $\leq 0$ holds. When $z^{(k)} := x^{(k+1)}$, (err) becomes nonpositive without step size constraints.

Or, since using the scheme we can write

$$\left\|x^{(k+1)} - z^{(k)}\right\|^2 = \left\|\frac{A_{k+1} - A_k}{A_{k+1}}v^{(k+1)} + \frac{A_k}{A_{k+1}}x^{(k)} - z^{(k)}\right\|^2,$$

by setting

$$z^{(k)} := \frac{A_{k+1} - A_k}{A_{k+1}}v^{(k)} + \frac{A_k}{A_{k+1}}x^{(k)},$$

we obtain

$$h \times \text{(err)} = \left(\frac{(A_{k+1} - A_k)^2}{A_{k+1}}\alpha - 2\right)\left\|v^{(k+1)} - v^{(k)}\right\|^2.$$

This choice of $z^{(k)}$ is shown in the theorem. When, for example, $A_k = (kh)^2$, (err) $\leq 0$, provided that $h \leq 1/\sqrt{2\alpha}$. Here we see that only up to a quadratic function is allowed as $A_k$ if $\alpha > 0$. ∎

### E.4 Proof of Theorem 5.5

It is sufficient to show that

$$\tilde{E}^{(k)} := f\left(x^{(k)}\right) - f^\star + (\beta + \gamma)\left\|v^{(k)} - x^\star\right\|^2$$

satisfies $\delta^+\tilde{E}^{(k)} \leq -\sqrt{m}\tilde{E}^{(k+1)}$. To simplify notation, $2(\beta+\gamma)$ is written as $m$, and the error terms are gathered into (err). Then, we see

$$\delta^+\tilde{E}^{(k)} = \delta^+ f\left(x^{(k)}\right) + \frac{m}{2}\delta^+\left\|v^{(k)} - x^\star\right\|^2$$

$$\leq \left\langle \overline{\nabla} f\left(x^{(k+1)}, z^{(k)}\right), \delta^+ x^{(k)} \right\rangle + \frac{\alpha}{h}\left\|x^{(k+1)} - z^{(k)}\right\|^2 - \frac{\beta}{h}\left\|z^{(k)} - x^{(k)}\right\|^2 - \gamma h\left\|\delta^+ x^{(k)}\right\|^2$$

$$+ m\left\langle \delta^+ v^{(k)}, v^{(k+1)} - x^\star \right\rangle - \frac{m}{2}h\left\|\delta^+ v^{(k)}\right\|^2$$

$$= \left\langle \overline{\nabla} f\left(x^{(k+1)}, z^{(k)}\right), \sqrt{m}\left(v^{(k+1)} - x^{(k+1)}\right) \right\rangle + (\text{err})$$

$$+ m\left\langle \sqrt{m}\left(\frac{2\beta}{m}z^{(k)} + \frac{2\gamma}{m}x^{(k+1)} - v^{(k+1)} - \frac{1}{m}\overline{\nabla} f\left(x^{(k+1)}, z^{(k)}\right)\right), v^{(k+1)} - x^\star \right\rangle$$

$$= \sqrt{m}\left\langle \overline{\nabla} f\left(x^{(k+1)}, z^{(k)}\right), x^\star - x^{(k+1)} \right\rangle - 2\sqrt{m}\beta\left\langle v^{(k+1)} - z^{(k)}, v^{(k+1)} - x^\star \right\rangle$$

$$- 2\sqrt{m}\gamma\left\langle v^{(k+1)} - x^{(k+1)}, v^{(k+1)} - x^\star \right\rangle + (\text{err})$$

$$= \sqrt{m}\left\langle \overline{\nabla} f\left(x^{(k+1)}, z^{(k)}\right), x^\star - x^{(k+1)} \right\rangle - \sqrt{m}\beta\left(\left\|v^{(k+1)} - z^{(k)}\right\|^2 + \left\|v^{(k+1)} - x^\star\right\|^2 - \left\|z^{(k)} - x^\star\right\|^2\right)$$

$$- \sqrt{m}\gamma\left(\left\|v^{(k+1)} - x^{(k+1)}\right\|^2 + \left\|v^{(k+1)} - x^\star\right\|^2 - \left\|x^{(k+1)} - x^\star\right\|^2\right) + (\text{err})$$

$$\leq -\sqrt{m}\left(f\left(x^{(k+1)}\right) - f^\star + \frac{m}{2}\left\|v^{(k+1)} - x^\star\right\|^2\right)$$

$$+ \sqrt{m}\left(\alpha\left\|x^{(k+1)} - z^{(k)}\right\|^2 - \beta\left\|v^{(k+1)} - z^{(k)}\right\|^2 - \gamma\left\|v^{(k+1)} - x^{(k+1)}\right\|^2\right) + (\text{err})$$

$$= -\sqrt{m}\tilde{E}^{(k+1)} + (\text{err}).$$

Here, the first inequality follows from (8) as the chain rule, the second equality from the substitution of the form of the method, and the second inequality follows from again (8) as the strongly convex inequality. In the second inequality, we also used

$$-\left\langle \overline{\nabla} f\left(x^{(k+1)}, z^{(k)}\right), x^{(k+1)} - x^\star \right\rangle + \beta\left\|z^{(k)} - x^\star\right\|^2 + \gamma\left\|x^{(k+1)} - x^\star\right\|^2 \leq -\left(f\left(x^{(k+1)}\right) - f^\star\right) + \alpha\left\|x^{(k+1)} - z^{(k)}\right\|^2.$$

Now we define $x^{(k)}$ so that $(\text{err}) \leq 0$. An obvious choice is $z^{(k)} := x^{(k+1)}$, where $(\text{err})$ is nonpositive under any step size.

To derive another definition of $z^{(k)}$, we proceed with the calculation of the error terms by substituting the form of the method:

$$h \times (\text{err}) = \alpha\left\|x^{(k+1)} - z^{(k)}\right\|^2 - \beta\left\|z^{(k)} - x^{(k)}\right\|^2 - \gamma\left\|x^{(k+1)} - x^{(k)}\right\|^2 - (\beta+\gamma)\left\|v^{(k+1)} - v^{(k)}\right\|^2$$

$$+ \sqrt{m}h\left(\alpha\left\|x^{(k+1)} - z^{(k)}\right\|^2 - \beta\left\|v^{(k+1)} - z^{(k)}\right\|^2 - \gamma\left\|v^{(k+1)} - x^{(k+1)}\right\|^2\right)$$

$$= \alpha(h+1)\left\|x^{(k+1)} - z^{(k)}\right\|^2 - \beta\left(\left\|z^{(k)} - x^{(k)}\right\|^2 + \left\|v^{(k+1)} - v^{(k)}\right\|^2 + \sqrt{m}h\left\|v^{(k+1)} - z^{(k)}\right\|^2\right)$$

$$- \gamma\left(\left\|x^{(k+1)} - x^{(k)}\right\|^2 + \left\|v^{(k+1)} - v^{(k)}\right\|^2 + \sqrt{m}h\left\|v^{(k+1)} - x^{(k+1)}\right\|^2\right)$$

$$= \alpha(h+1)\left\|x^{(k+1)} - z^{(k)}\right\|^2$$

$$- \beta\left(\left\|z^{(k)} - x^{(k)}\right\|^2 + \left\|v^{(k+1)} - v^{(k)}\right\|^2 + \sqrt{m}h\left\|x^{(k+1)} + \frac{x^{(k+1)} - x^{(k)}}{\sqrt{m}h} - z^{(k)}\right\|^2\right)$$

$$- \gamma\left(\left\|x^{(k+1)} - x^{(k)}\right\|^2 + \left\|v^{(k+1)} - v^{(k)}\right\|^2 + \sqrt{m}h\left\|x^{(k+1)} + \frac{x^{(k+1)} - x^{(k)}}{\sqrt{m}h} - x^{(k+1)}\right\|^2\right).$$

Hereafter, $\sqrt{m}h$ is denoted by $\tilde{h}$. By using Lemma F.2, we have

$$\tilde{h}\left\|x^{(k+1)} + \frac{x^{(k+1)} - x^{(k)}}{\tilde{h}} - z^{(k)}\right\|^2$$

$$= \tilde{h}\left(\frac{\tilde{h}+1}{\tilde{h}}\right)^2 \left\|\frac{\tilde{h}}{\tilde{h}+1}\left(x^{(k+1)} - z^{(k)}\right) + \frac{1}{\tilde{h}+1}\left(x^{(k+1)} - x^{(k)}\right)\right\|^2$$

$$= \frac{\left(\tilde{h}+1\right)^2}{\tilde{h}}\left(\frac{\tilde{h}}{\tilde{h}+1}\left\|x^{(k+1)} - z^{(k)}\right\|^2 + \frac{1}{\tilde{h}+1}\left\|x^{(k+1)} - x^{(k)}\right\|^2 - \frac{\tilde{h}}{\left(\tilde{h}+1\right)^2}\left\|z^{(k)} - x^{(k)}\right\|^2\right)$$

$$= \left(\tilde{h}+1\right)\left\|x^{(k+1)} - z^{(k)}\right\|^2 + \frac{\tilde{h}+1}{\tilde{h}}\left\|x^{(k+1)} - x^{(k)}\right\|^2 - \left\|z^{(k)} - x^{(k)}\right\|^2.$$

Thus, we see

$$\tilde{h} \times (\text{err}) = (\alpha - \beta)\left(\tilde{h}+1\right)\left\|x^{(k+1)} - z^{(k)}\right\|^2 - (\beta+\gamma)\left(\frac{\tilde{h}+1}{\tilde{h}}\left\|x^{(k+1)} - x^{(k)}\right\|^2 + \left\|v^{(k+1)} - v^{(k)}\right\|^2\right).$$

Here when we define $z^{(k)} := x^{(k)}$, (err) is nonpositive under the condition

$$(\alpha - \beta)\left(\tilde{h}+1\right) - (\beta+\gamma)\frac{\tilde{h}+1}{\tilde{h}} \leq 0.$$

This condition reads as $\tilde{h} \leq (\beta+\gamma)/(\alpha-\beta)$, and the convergence rate is

$$\left(1 + \sqrt{2(\beta+\gamma)}\right)^{-k} = \left(1 + \tilde{h}\right)^{-k} \geq \left(1 - \frac{\beta+\gamma}{\alpha+\gamma}\right)^k.$$

To obtain a better rate, we continue the computation of (err) without defining $z^{(k)}$. Let

$$\eta = \frac{1}{\frac{\tilde{h}+1}{\tilde{h}} + 1} = \frac{\tilde{h}}{2\tilde{h}+1},$$

and again by inserting the form of the method, and by using Lemma F.2,

$$\frac{\tilde{h}+1}{\tilde{h}}\left\|x^{(k+1)} - x^{(k)}\right\|^2 + \left\|v^{(k+1)} - v^{(k)}\right\|^2$$

$$= \frac{\tilde{h}+1}{\tilde{h}}\left\|x^{(k+1)} - x^{(k)}\right\|^2 + \left\|x^{(k+1)} + \frac{x^{(k+1)} - x^{(k)}}{\tilde{h}} - v^{(k)}\right\|^2$$

$$= \frac{\tilde{h}+1}{\tilde{h}}\left\|x^{(k+1)} - x^{(k)}\right\|^2 + \left(\frac{\tilde{h}+1}{\tilde{h}}\right)^2\left\|x^{(k+1)} - \frac{\tilde{h}}{\tilde{h}+1}v^{(k)} - \frac{1}{\tilde{h}+1}x^{(k)}\right\|^2$$

$$= \frac{\tilde{h}+1}{\tilde{h}}\frac{1}{\eta}\left(\eta\left\|x^{(k+1)} - x^{(k)}\right\|^2 + (1-\eta)\left\|x^{(k+1)} - \frac{\tilde{h}}{\tilde{h}+1}v^{(k)} - \frac{1}{\tilde{h}+1}x^{(k)}\right\|^2\right)$$

$$= \frac{\tilde{h}+1}{\tilde{h}}\frac{1}{\eta}\left(\left\|\eta\left(x^{(k+1)} - x^{(k)}\right) + (1-\eta)\left(x^{(k+1)} - \frac{\tilde{h}}{\tilde{h}+1}v^{(k)} - \frac{1}{\tilde{h}+1}x^{(k)}\right)\right\|^2\right.$$

$$\left. +\eta(1-\eta)\left\|x^{(k+1)} - x^{(k)} - \left(x^{(k+1)} - \frac{\tilde{h}}{\tilde{h}+1}v^{(k)} - \frac{1}{\tilde{h}+1}x^{(k)}\right)\right\|^2\right)$$

$$= \frac{\tilde{h}+1}{\tilde{h}}\frac{2\tilde{h}+1}{\tilde{h}}\left[\left\|x^{(k+1)} - \left(\frac{\tilde{h}+1}{2\tilde{h}+1}x^{(k)} + \frac{\tilde{h}}{2\tilde{h}+1}v^{(k)}\right)\right\|^2 + \frac{\tilde{h}}{2\tilde{h}+1}\frac{\tilde{h}+1}{2\tilde{h}+1}\left(\frac{\tilde{h}}{\tilde{h}+1}\right)^2\left\|v^{(k)} - x^{(k)}\right\|^2\right].$$

Hence we obtain

$$\tilde{h} \times (\text{err}) = (\alpha - \beta)\left(\tilde{h} + 1\right)\left\|x^{(k+1)} - z^{(k)}\right\|^2 - (\beta + \gamma)\frac{\tilde{h}}{2\tilde{h} + 1}\left\|v^{(k)} - x^{(k)}\right\|^2$$

$$- (\beta + \gamma)\frac{\tilde{h} + 1}{\tilde{h}}\frac{2\tilde{h} + 1}{\tilde{h}}\left\|x^{(k+1)} - \frac{\tilde{h} + 1}{2\tilde{h} + 1}x^{(k)} - \frac{\tilde{h}}{2\tilde{h} + 1}v^{(k)}\right\|^2.$$

If we set

$$z^{(k)} := \frac{\tilde{h} + 1}{2\tilde{h} + 1}x^{(k)} + \frac{\tilde{h}}{2\tilde{h} + 1}v^{(k)},$$

(err) is nonpositive under the condition

$$(\alpha - \beta)\left(\tilde{h} + 1\right) - (\beta + \gamma)\frac{\tilde{h} + 1}{\tilde{h}}\frac{2\tilde{h} + 1}{\tilde{h}} \le 0.$$

The definition of $z^{(k)}$ is shown in the theorem. By solving the above inequality, we obtain the step size limitation

$$\tilde{h} \le \frac{\sqrt{\beta + \gamma}}{\sqrt{\alpha + \gamma} - \sqrt{\beta + \gamma}},$$

which is shown in the theorem. ∎

## F  Law of cosines and parallelogram identity

This section summarizes some useful lemmas used in the preceding sections.

In Hilbert spaces, especially in Euclidean spaces, the law of cosines holds:

$$\|y - x\|^2 = \|y\|^2 + \|x\|^2 - 2\langle y, x\rangle.$$

In this paper, we use this formula as an error-containing discrete chain rule of the squared norm.

**Lemma F.1.** *For all $x^{(k+1)}, x^{(k)} \in \mathbb{R}^d$,*

$$\left\|x^{(k+1)}\right\|^2 - \left\|x^{(k)}\right\|^2 = 2\langle x^{(k+1)}, x^{(k+1)} - x^{(k)}\rangle - \left\|x^{(k+1)} - x^{(k)}\right\|^2$$

Another famous identity for the Hilbert norm (especially the Euclidean norm) is the parallelogram identity:

$$\left\|\frac{x + y}{2}\right\|^2 + \left\|\frac{x - y}{2}\right\|^2 = \frac{1}{2}(\|x\|^2 + \|y\|^2).$$

In this paper, we use a generalization of this identity. In the following lemma, we recover the parallelogram identity by setting $\alpha = 1/2$.

**Lemma F.2.** *For all $x, y \in \mathbb{R}^d$ and $\alpha \in \mathbb{R}$,*

$$\|\alpha x + (1 - \alpha)y\|^2 = \alpha\|x\|^2 + (1 - \alpha)\|y\|^2 - \alpha(1 - \alpha)\|x - y\|^2.$$

*Proof.* The claim is obtained by adding each side of the following equalities:

$$\|\alpha x + (1 - \alpha)y\|^2 = \alpha^2\|x\|^2 + (1 - \alpha)^2\|y\|^2 + 2\alpha(1 - \alpha)\langle x, y\rangle,$$
$$\alpha(1 - \alpha)\|x - y\|^2 = \alpha(1 - \alpha)\|x\|^2 + \alpha(1 - \alpha)\|y\|^2 - 2\alpha(1 - \alpha)\langle x, y\rangle.$$

∎

## G  Proofs of theorems in Section 6

### G.1  Proof of Theorem H.1

It is sufficient to show that

$$E(t) := f(x(t)) - f^\star$$

satisfies $\dot{E} \le -2\mu E$. Indeed,

$$\dot{E} = \langle \nabla f(x), \dot{x}\rangle = -\|\nabla f(x)\|^2 \le -2\mu(f(x) - f^\star) = -2\mu E$$

holds. Here, we used the chain rule, the continuous system itself, the PŁ condition, and the definition of $E$ in this order. ∎

## G.2 Proof of Theorem H.3

By the PŁ condition, we observe that

$$-\left\|\overline{\nabla}f(y,x)\right\| \leq -\sqrt{2\mu(f(x)-f^\star)} + \|\nabla f(x)\| - \left\|\overline{\nabla}f(y,x)\right\|$$
$$\leq -\sqrt{2\mu(f(x)-f^\star)} + \left\|\nabla f(x) - \overline{\nabla}f(y,x)\right\|.$$

Thus, the evaluation of $\left\|\overline{\nabla}f(y,x) - \nabla f(x)\right\|$ yields $\beta$.

(i) From $L$-smoothness $\alpha = L/2$ follows. By the definition $\beta = 0$.

(ii) $L$-smoothness yields $\alpha = L/2$ and $\beta = L$. (Note that the convexity of $f$ would imply $\alpha = 0$, but it is not assumed now. If we adopt the other definition (18) then $\beta = 0$.)

(iii) By the same application of $L$-smoothness, we obtain $\alpha = L/8$ and $\beta = L/2$.

(iv) Since the discrete chain rule exactly holds, $\nabla_{\mathrm{AVF}}f$ satisfies $\alpha = 0$. Then, by the $L$-smoothness of $f$,

$$\begin{aligned}
\|\nabla_{\mathrm{AVF}}f(y,x) - \nabla f(x)\| &= \left\|\int_0^1 \nabla f(\tau y + (1-\tau)x)\mathrm{d}\tau - \nabla f(x)\right\| \\
&\leq \int_0^1 \|\nabla f(\tau y + (1-\tau)x) - \nabla f(x)\|\mathrm{d}\tau \\
&\leq \int_0^1 L\|\tau y + (1-\tau)x - x\|\mathrm{d}\tau \\
&\leq \int_0^1 L\tau\|y - x\|\mathrm{d}\tau \\
&\leq \frac{L}{2}\|y - x\|
\end{aligned}$$

holds, which implies $\beta = L/2$.

(v) Similar to the case (iv), $\alpha = 0$ holds. By the $L$-smoothness of $f$,

$$\begin{aligned}
\|\nabla_{\mathrm{G}}f(y,x) - \nabla f(x)\| &= \left\|\nabla f\left(\frac{y+x}{2}\right) - \frac{f(y) - f(x) - \left\langle\nabla f\left(\frac{y+x}{2}\right), y-x\right\rangle}{\|y-x\|^2}(y-x) - \nabla f(x)\right\| \\
&\leq \left\|\nabla f\left(\frac{y+x}{2}\right) - \nabla f(x)\right\| + \frac{\left|f(y) - f(x) - \left\langle\nabla f\left(\frac{y+x}{2}\right), y-x\right\rangle\right|}{\|y-x\|} \\
&\leq \frac{L}{2}\|y-x\| + \frac{L}{8}\|y-x\| \\
&= \frac{5L}{8}\|y-x\|,
\end{aligned}$$

which implies $\beta = 5L/8$.

(vi) Similar to the previous cases, $\alpha = 0$ holds. Using the same notation as in Appendix D (vi), we obtain

$$\|\nabla_{\mathrm{IA}} f(y, x) - \nabla f(x)\| = \left\| \begin{bmatrix} \frac{f(y_1, x_2, x_3 \ldots, x_d) - f(x_1, x_2, x_3, \ldots, x_d)}{y_1 - x_1} \\ \frac{f(y_1, y_2, x_3 \ldots, x_d) - f(y_1, x_2, x_3, \ldots, x_d)}{y_2 - x_2} \\ \vdots \\ \frac{f(y_1, y_2, y_3, \ldots, y_d) - f(y_1, y_2, y_3 \ldots, x_d)}{y_d - x_d} \end{bmatrix} - \nabla f(x) \right\|$$

$$= \left\| \begin{bmatrix} \partial_1 f(\theta_1 y_1 + (1 - \theta_1)x_1, x_2, x_3 \ldots, x_d) \\ \partial_2 f(y_1, \theta_2 y_2 + (1 - \theta_2)x_2, x_3 \ldots, x_d) \\ \vdots \\ \partial_d f(y_1, y_2, y_3, \ldots, \theta_d y_d + (1 - \theta_d)x_d) \end{bmatrix} - \nabla f(x) \right\|$$

$$= \sqrt{\sum_{k=1}^{d} |(\nabla f(\theta_k y_{1:k} x_{k+1:d} + (1 - \theta_k)y_{1:k-1} x_{k:d}))_k - (\nabla f(x))_k|^2}$$

$$\leq \sqrt{\sum_{k=1}^{d} L^2 \|y - x\|^2}$$

$$= \sqrt{d} L \|y - x\|,$$

where $\theta_k \in [0, 1]$ is a constant by the mean value theorem. Therefore, $\beta = \sqrt{d} L$ holds.

∎

### G.3 Proof of Theorem H.5

Let

$$\tilde{E}^{(k)} := f\left(x^{(k)}\right) - f^\star.$$

If $\delta^+ \tilde{E}^{(k)} \leq -c\tilde{E}^{(k)}$ for $c > 0$, it can be concluded that $E^{(k)} = (1 - ch)^{-k} \tilde{E}^{(k)}$ is nonincreasing and hence $f\left(x^{(k)}\right) - f^\star \leq (1 - ch)^k E^{(0)}$. Before starting the computation of $\delta^+ \tilde{E}^{(k)}$, we transform the weak discrete PŁ condition (17) into a more convenient form. By substituting the scheme into (17), we obtain

$$-\left\| \overline{\nabla} f(x^{(k+1)}, x^{(k)}) \right\|^2 \leq -\frac{\gamma}{(1 + \beta h)^2} \left( f\left(x^{(k)}\right) - f^\star \right).$$

Thus, it follows from the weak discrete chain rule (16), the scheme, and the above inequality, that

$$\delta^+ \tilde{E}^{(k)} = \delta^+ f\left(x^{(k)}\right)$$

$$\leq \left\langle \overline{\nabla} f\left(x^{(k+1)}, x^{(k)}\right), \delta^+ x^{(k)} \right\rangle + \alpha h \left\| \delta^+ x^{(k)} \right\|^2$$

$$= -(1 - \alpha h) \left\| \overline{\nabla} f\left(x^{(k+1)}, x^{(k)}\right) \right\|^2$$

$$\leq -(1 - \alpha h) \frac{\gamma}{(1 + \beta h)^2} \left( f\left(x^{(k)}\right) - f^\star \right)$$

$$= -\gamma \frac{1 - \alpha h}{(1 + \beta h)^2} E^{(k)}.$$

Hence if $h \leq 1/\alpha$ we have the convergence.

∎

## H  Extension to Polyak–Łojasiewicz type functions.

The weak discrete gradients can be useful also for non-convex functions. Here we illustrate it by taking functions satisfying the Polylak–Łojasiewicz (PŁ) condition. A function $f$ is said to satisfy the PŁ condition if

$$-\|\nabla f(x)\|^2 \leq -2\mu(f(x) - f^\star)$$

holds for any $x \in \mathbb{R}^d$. This was introduced as a sufficient condition for the steepest descent to converge Polyak (1963). The set of functions satisfying the PŁ condition contains all differentiable strongly convex functions and some nonconvex functions such as $f(x) = x^2 + 3\sin^2(x)$.

**Theorem H.1** (Continuous systems). *Suppose that $f$ satisfies the PŁ condition. Let $x\colon [0, \infty) \to \mathbb{R}^d$ be the solution of the gradient flow (3). Then the solution satisfies*

$$f(x(t)) - f^\star \leq \mathrm{e}^{-\mu t} \|x_0 - x^\star\|^2.$$

Let us define another weak discrete gradient for functions satisfying the PŁ condition. Recall that the first condition of weak discrete gradients (Definition 4.1) has two meanings: the discrete chain rule (9) and the discrete convex inequality (10). One could consider the PŁ condition instead of convexity.

**Definition H.2.** A gradient approximation $\overline{\nabla} f \colon \mathbb{R}^d \times \mathbb{R}^d \to \mathbb{R}^d$ is said to be *PŁ-type weak discrete gradient of $f$* if there exists positive numbers $\alpha, \beta$ such that for all $x, y \in \mathbb{R}^d$ the following three conditions hold:

$$\overline{\nabla} f(x, x) = \nabla f(x),$$
$$f(y) - f(x) \leq \langle \overline{\nabla} f(y, x), y - x \rangle + \alpha \|y - x\|^2, \tag{16}$$
$$-\left\|\overline{\nabla} f(y, x)\right\| \leq -\sqrt{2\mu(f(x) - f^\star)} + \beta \|y - x\|. \tag{17}$$

**Theorem H.3.** *If $f$ is $L$-smooth and satisfies the PŁ condition with the parameter $\mu$, the following functions are PŁ-type weak discrete gradients:*

(i) *If $\overline{\nabla} f(y, x) = \nabla f(x)$; then $(\alpha, \beta) = (L/2, 0)$.*

(ii) *If $\overline{\nabla} f(y, x) = \nabla f(y)$, then $(\alpha, \beta) = (L/2, L)$.*

(iii) *If $\overline{\nabla} f(y, x) = \nabla f(\frac{x+y}{2})$, then $(\alpha, \beta) = (L/8, L/2)$.*

(iv) *If $\overline{\nabla} f(y, x) = \nabla_{\mathrm{AVF}} f(y, x)$, then $(\alpha, \beta) = (0, L/2)$.*

(v) *If $\overline{\nabla} f(y, x) = \nabla_{\mathrm{G}} f(y, x)$, then $(\alpha, \beta) = (0, 5L/8)$.*

(vi) *If $\overline{\nabla} f(y, x) = \nabla_{\mathrm{IA}} f(y, x)$, then $(\alpha, \beta) = (0, \sqrt{d}L)$.*

*Remark* H.4. The parameters $\alpha$ and $\beta$ imply the magnitude of the discretization error. As the second condition in Definition H.2, we can adopt instead

$$-\left\|\overline{\nabla} f(y, x)\right\| \leq -\sqrt{2\mu(f(y) - f^\star)} + \beta \|y - x\|. \tag{18}$$

Then, we obtain better parameters for the implicit Euler method (ii).

The proof of Theorem H.3 is postponed in Appendix G.2.

**Theorem H.5** (Discrete systems). *Let $\overline{\nabla} f$ be a PŁ-type weak discrete gradient of $f$. Let $f$ be a function which satisfies the necessary conditions that the PŁ-type weak DG requires. Let $\{x^{(k)}\}$ be the sequence given by (11). Then, under the step size condition $h \leq 1/\alpha$, the sequence satisfies*

$$f\left(x^{(k)}\right) - f^\star \leq \left(1 - 2\mu h \frac{(1 - \alpha h)}{(1 + \beta h)^2}\right)^k (f(x_0) - f^\star).$$

*In particular, the sequence satisfies*

$$f\left(x^{(k)}\right) - f^\star \leq \left(1 - \frac{\mu}{2(\alpha + \beta)}\right)^k (f(x_0) - f^\star),$$

*when the optimal step size $h = 1/(2\alpha + \beta)$ is employed.*

## I  Some numerical examples

In this section, we give some numerical examples to complement the discussion in the main body of this paper. Note that this is just to illustrate that we can actually easily construct new concrete

methods just by assuring the conditions of weak discrete gradients (weak DGs), and that the resulting methods in fact achieve the prescribed rates; we here do not intend to explore a method that beats known state-of-the-art methods. It is of course an ultimate goal of the unified framework project, but is left as an important future work.

Below we consider some explicit optimization methods derived as special cases of the abstract weak DG methods. Here we pick up simple two-dimensional problems so that we can observe not only the decrease of the objective functions but also the trajectories of the points $x$'s for our intuitive understandings.

First, we consider the case where the objective function is a $L$-smooth convex function. An explicit weak discrete gradient method is then found as

$$\begin{cases} x^{(k+1)} - x^{(k)} = \dfrac{2k+1}{k^2}\Big(v^{(k+1)} - x^{(k+1)}\Big), \\[2mm] v^{(k+1)} - v^{(k)} = -\dfrac{2k+1}{4}h^2\nabla f\Big(z^{(k)}\Big), \\[2mm] z^{(k)} - x^{(k)} = \dfrac{2k+1}{(k+1)^2}\Big(v^{(k)} - x^{(k)}\Big). \end{cases} \tag{19}$$

We call this method (wDG-c). This is the simplest example of the abstract method in Theorem 5.4, where we choose $A_k = (kh)^2$ and $\overline{\nabla} f(y, x) = \nabla f(x)$. The authors believe this method itself has not been explicitly pointed out in the literature, and is new. The expected rate is the one predicted in the theorem, $\mathrm{O}\big(1/k^2\big)$, under the step size condition $h \le 1/\sqrt{L}$ (recall that $\alpha$ for the weak DG is $L/2$ as shown in Theorem 4.2). For comparison, we pick up Nesterov's accelerated gradient method for convex functions

$$\begin{cases} y^{(k+1)} = x^{(k)} - h^2\nabla f\Big(x^{(k)}\Big), \\[2mm] x^{(k+1)} = y^{(k+1)} + \dfrac{k}{k+3}\Big(y^{(k+1)} - y^{(k)}\Big). \end{cases} \tag{20}$$

We denote this method by (NAG-c). It is well-known that it achieves the same rate, under the same step size condition; we summarize these information in Table 3.

As an objective function, we employ

$$f(x) = 0.1x_1{}^4 + 0.001x_2{}^4, \tag{21}$$

which is not strongly convex. (Strictly speaking, this is not $L$-smooth as well, but we consider in the following way: for each initial $x$ we obtain the level set $\{x \mid f(x) = f(x^{(0)})\}$. Then we consider the function (21) inside the region, and extend the function outside it appropriately; for example such that the function grows linearly as $\|x\| \to \infty$.)

Numerical results are shown in Figure 1. The top-left panel of the figure shows the convergence of the objective function $f(x)$ when the optimal step size $1/\sqrt{L}$ is chosen. We see both methods achieve the predicted rate $\mathrm{O}\big(1/k^2\big)$ (mind the dotted guide line). We also see that under this setting, (wDG-c) converges faster than (NAG-c). This suggests that the new framework can give rise to an optimization method that is competitive to state-of-the-art methods (as said before, we do not say anything conclusive on this point; in order to discuss practical performance, we further need to discuss other implementation issues such as stepping schemes.) The trajectories of $x^{(k)}$'s are almost the same, for all the tested step sizes.

Next, we consider methods for strongly convex functions. We use the following explicit weak DG method:

$$\begin{cases} x^{(k+1)} - x^{(k)} = \sqrt{\mu}h\Big(v^{(k+1)} - x^{(k+1)}\Big), \\[2mm] v^{(k+1)} - v^{(k)} = \sqrt{\mu}h\left(z^{(k)} - v^{(k+1)} - \dfrac{\nabla f\big(z^{(k)}\big)}{\mu}\right), \\[2mm] z^{(k)} - x^{(k)} = \sqrt{\mu}h\Big(x^{(k)} + v^{(k)} - 2z^{(k)}\Big). \end{cases} \tag{22}$$

We call this (wDG-sc). This can be obtained by setting $\overline{\nabla} f(y, x) = \nabla f(x)$ in Theorem 5.5. Since for this choice we have $\alpha = L/2$ and $\beta = \mu/2$ (Theorem 4.2), the step size condition is

Table 3: Step size limitations and convergence rates of the methods used in the experiments

| scheme | step size limitation | convergence rate |
|---|---|---|
| NAG-c (20) | $1/\sqrt{L}$ | $O\left(\frac{1}{k^2}\right)$ |
| wDG-c (19) | $1/\sqrt{L}$ | $O\left(\frac{1}{k^2}\right)$ |
| NAG-sc (23) | $1/\sqrt{L}$ | $O\left(\left(1-\sqrt{\frac{\mu}{L}}\right)^k\right)$ |
| wDG-sc (22) | $1/(\sqrt{L}-\sqrt{\mu})$ | $O\left(\left(1-\sqrt{\frac{\mu}{L}}\right)^k\right)$ |
| wDG2-sc (24) | $\sqrt{\mu}/(L-\mu)$ | $O\left(\left(1-\frac{\mu}{L}\right)^k\right)$ |

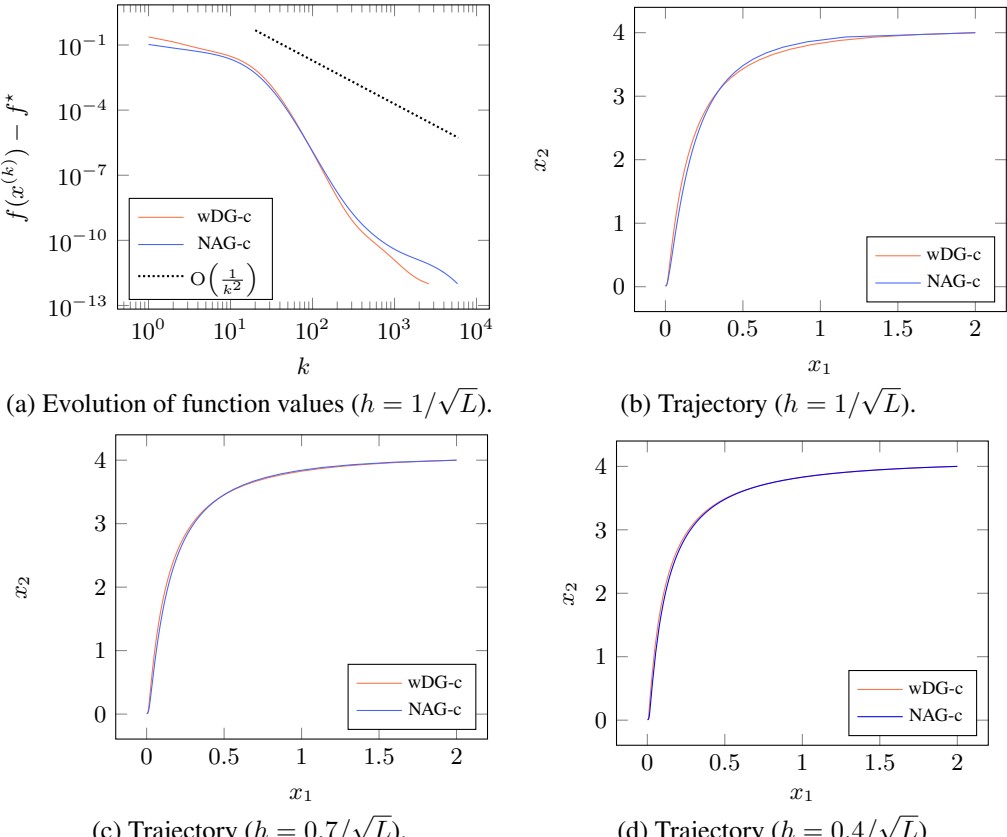

(a) Evolution of function values ($h = 1/\sqrt{L}$).

(b) Trajectory ($h = 1/\sqrt{L}$).

(c) Trajectory ($h = 0.7/\sqrt{L}$).

(d) Trajectory ($h = 0.4/\sqrt{L}$)

Figure 1: Trajectories and function values by (wDG-c) and (NAG-c). The objective function is (21) and the initial solution is $(2, 4)$.

$h \leq 1/(\sqrt{L} - \sqrt{\mu})$, and the predicted rate is $\mathrm{O}\left((1 - \sqrt{\mu/L})^k\right)$ (which is attained by the largest $h = 1/(\sqrt{L} - \sqrt{\mu})$).

As before, we compare this method with Nesterov's accelerated gradient method for strongly convex functions (NAG-sc):

$$
\begin{cases}
y^{(k+1)} = x^{(k)} - h^2 \nabla f\left(x^{(k)}\right), \\
x^{(k+1)} = y^{(k+1)} + \dfrac{1 - \sqrt{\mu}h}{1 + \sqrt{\mu}h}\left(y^{(k+1)} - y^{(k)}\right).
\end{cases}
\tag{23}
$$

Here, in addition to these, we also consider a simpler method, where $z^{(k)} = x^{(k)}$ is chosen to find

$$
\begin{cases}
x^{(k+1)} - x^{(k)} = \sqrt{\mu}h\left(v^{(k+1)} - x^{(k+1)}\right), \\
v^{(k+1)} - v^{(k)} = \sqrt{\mu}h\left(x^{(k)} - v^{(k+1)} - \dfrac{\nabla f\left(x^{(k)}\right)}{\mu}\right).
\end{cases}
\tag{24}
$$

We call this (wDG2-sc). This method is more natural as a numerical method for the accelerated gradient flow (6), compared to the methods above, and we expect it illustrates how "being natural as a numerical method" affects the performance. The rate and the step size limitation were revealed in the proof of Theorem 5.5 (Appendix E).

We summarized the step size limitations and rates in Table 3. Notice that the predicted rate of (wDG-sc) is better than that of (wDG2-sc).

The objective function is taken to be the quadratic function

$$
f(x) = 0.001(x_1 - x_2)^2 + 0.1(x_1 + x_2)^2 + 0.01x_1 + 0.02x_2,
\tag{25}
$$

and results are shown is Figure 2. Again the top-left panel shows the convergence of the objective function. We see that (wDG-sc) and (NAG-sc) with each optimal step size show almost the same convergence, which is in this case much better than the predicted worst case rate (the dotted guide line). (wDG2-sc) slightly falls behind the other two, but it eventually achieves almost the same performance as $k \to \infty$. The trajectories of the points $x^{(k)}$'s are, however, quite different among the three methods, which is interesting to observe. The trajectory of (wDG2-sc) seems to suffer from wild oscillations, while (wDG-sc) generates milder trajectory. (NAG-sc) comes between these two. We need careful discussion to conclude which dynamics is the best as an optimization method, but if we consider such oscillations are not desirable (possibly causing some instability), it might suggest that (wDG-sc) is the first choice for this problem. In any case, in this way we can explore various concrete optimization method within the framework of the weak DG by varying the weak DG, which is exactly the main claim of this paper.

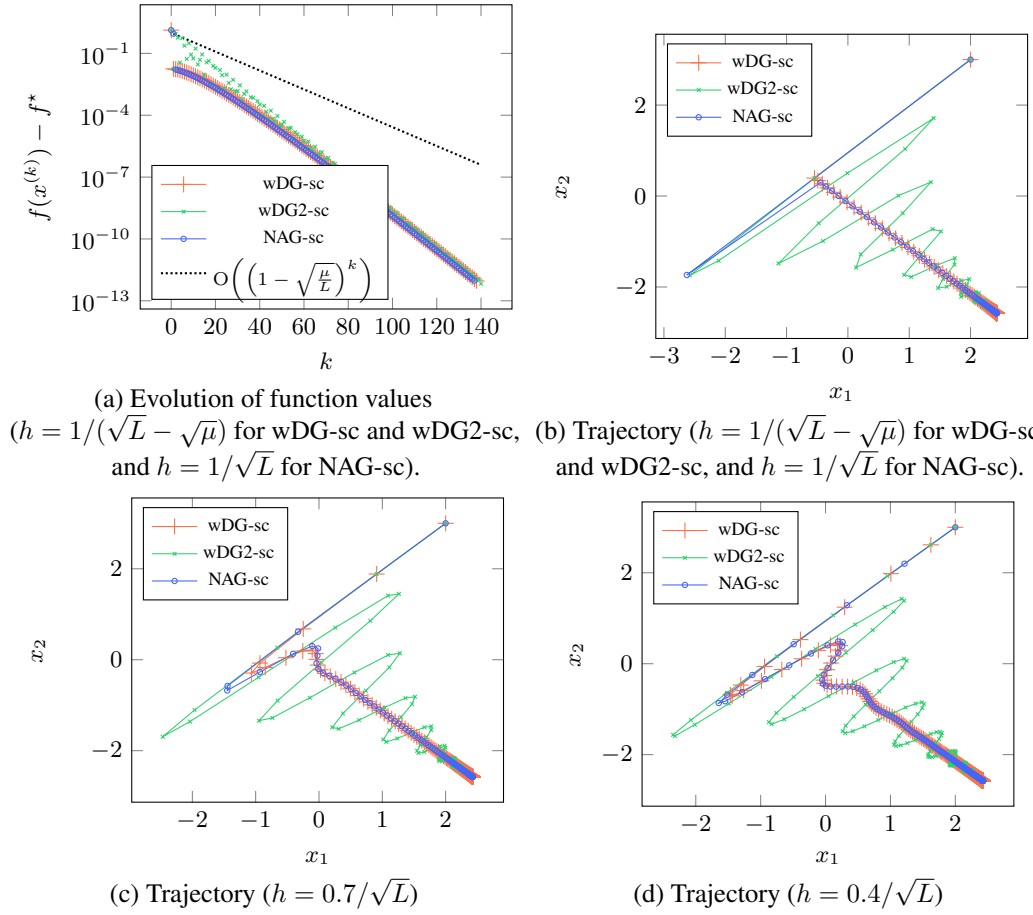

(a) Evolution of function values ($h = 1/(\sqrt{L} - \sqrt{\mu})$ for wDG-sc and wDG2-sc, and $h = 1/\sqrt{L}$ for NAG-sc).

(b) Trajectory ($h = 1/(\sqrt{L} - \sqrt{\mu})$ for wDG-sc and wDG2-sc, and $h = 1/\sqrt{L}$ for NAG-sc).

(c) Trajectory ($h = 0.7/\sqrt{L}$)

(d) Trajectory ($h = 0.4/\sqrt{L}$)

Figure 2: Trajectories and function values by (wDG-sc), (wDG2-sc) and (NAG-sc). The objective function is (25) and the initial value is $(2, 3)$.

