# OpenReview forum: "A Unified Discretization Framework for Differential Equation Approach with Lyapunov Arguments for Convex Optimization"
_NeurIPS.cc/2023/Conference — NeurIPS 2023 poster_

### Official Review · Reviewer_7m3u · 2023-06-21

**Soundness:** 4 excellent
**Presentation:** 4 excellent
**Contribution:** 2 fair
**Rating:** 5
**Confidence:** 3

**Summary:**

The paper proposes a unified discretization framework for the differential equation (DE) approach to convex optimization. The DE approach relates optimization methods to continuous DEs with Lyapunov functionals, providing intuitive insights and convergence rate estimates. However, there has been a lack of a general and consistent way to transition back to discrete optimization methods. The paper introduces the concept of "weak discrete gradient" (wDG), which consolidates the conditions required for discrete versions of gradients in the DE approach arguments. The authors define abstract optimization methods using wDG and demonstrate that many existing optimization methods and their convergence rates can be derived as special cases of their setting.

**Strengths:**

Comprehensive Approach: The paper addresses an important gap in the DE approach to convex optimization by providing a unified discretization framework. It consolidates the conditions required for discrete versions of gradients, simplifying the overall analysis.

Clarity: The pedagogical approach of the paper is appreciated. Most of the theory is constructive and all new aspect are clearly presented.

Abstract Optimization Methods: The introduction of abstract optimization methods using wDG allows for a simpler view and analysis of existing optimization methods. It provides a systematic framework for deriving convergence rate estimates.

Potential for new methods: The framework allows for the development of new optimization methods by combining known DEs with wDG. It opens up opportunities for researchers to explore novel approaches in the optimization field.

**Weaknesses:**

Knowing the accelerated gradient flow: The biggest problem of the approach presented in this paper is the requirement that one needs to know the accelerated gradient flow to derive its discretization - accelerated gradient flow that has been derived from an accelerated algorithm. Hence, the practicability of the approach is limited as one requires to know already an accelerated method before deriving another one.

Case-Specific Discrete Arguments: The authors acknowledge the need for adjustment and optimization in the construction of optimization methods: while the framework consolidates the case-specific discrete arguments in the definition of wDG, there may still be some complexity involved in finding new pratical wDGs for different methods.

**Questions:**

- Can the framework be easily extended to handle constrained optimizations (for instance, using indicator functions)?

- Theorem 4.2 list several wDG for smooth and/or strongly convex function. Is it possible to find easily other wDG for sunction that does not satisfies those assumptions? (for instance, non convex smooth functions, or functions with smooth second-order derivatives?)

- Can the limitations of the framework, such as the non-optimality of certain convergence rates, be overcome by considering alternative DEs and Lyapunov functionals?

**Limitations:**

- Non-Universal Applicability: The paper acknowledges that certain optimization methods, such as the Douglas-Rachford splitting method and methods based on Runge-Kutta numerical methods, may not fit into the current framework. The applicability of wDG to these methods remains an open question.

- Practicability: While the paper indeed overcome some limitation from previous work in this vein, given the previous points in the Weakness section, it seems the approach overcomplicate the design of new accelerated methods in other settings.

---

> ### Author Rebuttal · Authors · 2023-08-04
>
> Thank you very much for your careful reading and suggestions. We really appreciate it.
> Below are our responses to your criticisms in Weaknesses and Questions.
>
> >Weakness 1. Knowing the accelerated gradient flow:
>
> We feel the reviewer misunderstood our problem setting. Let us explain our standpoint again as follows.
> It is surely true that our framework is on how we discretize ODEs in the ODE approach, and it cannot be invoked if we do not have any ODEs (and Lyapunov functionals). But this does not mean that we have to have a new (discrete) optimization method first to start a new generalization;  instead one can start with freely exploiting new ODEs with rate-revealing Lyapunov functionals (In fact, Celine--Taylor--Bach (2022, arXiv:2205.12772) derived ODE and Lyapunov functional without basing on existing optimization methods, and Kim--Yang (2023, ICML2023) integrated known accelerated gradient flows to derive a new ODE that has good natures of both). Then if the ODE only uses gradients (i.e. does not involve Hessian or higher derivatives), it should be an easy task to define an abstract wDG method for it and write down a discrete convergence estimate just by following the proof in the continuous case (as we demonstrated in the present paper for known ODEs/Lyapunov functionals).
> In this sense, we claim that now people can almost forget about the final discretizations, and purely concentrate on the exploration of new ODEs/Lyapunov functionals. That is exactly the best outcome of our framework,  originally hoped in the seminal work by Su--Boyd--Candes (2014). These explanations were given in L62--69 (although it was a bit hasty due to the limitation of the space.)
>
> >Weakness 2.  Case-Specific Discrete Arguments:
>
> We feel there is some confusion again in the reviewer. What we meant by "Although there is still room for minor adjustments"(L72) is described in L303--309; roughly speaking, the choice of some additional variables, and/or optimization of time-stepping. These rooms are not necessarily negative, and in some case we can utilize them for further optimizing resulting optimization methods. Even if we leave these freedoms to some obvious choices, the framework gives a convergent method and its rate.
>
> We agree with the reviewer's view that finding a new, good wDG would be a nontrivial task. But this could be done as in usual research on creating new optimization methods; suppose we are considering some modification of an existing first-order method. Then, in our framework, it suffices to find out what serves as a wDG there, and check if it satisfies the condition of being a wDG. If so, then we know the convergence and the rate. It is also a good idea to start with an existing wDG, and consider its modification, keeping the conditions for wDG satisfied. In these ways, we believe our framework can serve as a good working environment also for considering new discretizations.
> These were briefly explained in L62--69.
>
> >Question 1.  Can the framework be easily extended to handle constrained optimizations (for instance, using indicator functions)?
>
> Yes. At this moment we have at least two directions for achieving this.
> The first direction is to follow the DE approach for mirror descent methods, which is very briefly mentioned in Rem 1.1.
> The second direction is to consider including indicating functions, as suggested by the reviewer. Because a linear combination of wDGs is again a wDG, we can consider a new objective function $\tilde{f} = f + \delta_C$ (where $\delta_C$ is the indicating function for the domain $C$) and their wDGs.
> These topics are outside the scope of this paper (due to the space restriction), but we have already confirmed that they work, and will report somewhere in the near future.
>
> >Question 2. Theorem 4.2 list several wDG for smooth and/or strongly convex function. Is it possible to find easily other wDG for function that does not satisfies those assumptions? (for instance, non convex smooth functions, or functions with smooth second-order derivatives?)
>
> We cordially ask the reviewer to distinguish problems in our framework and those in the entire DE approach.
> For non-convex and/or non-smooth functions, difficulties first lie in the DE approach itself; extensions to these objective functions are hoped, but are still left to be investigated, as far as the present authors understand.
> For functions with higher smoothness, investigations should be done both in the DE approach and our framework. As far as we know, no DE approach theory has been established for such objective functions. Even if it does, the authors are not sure if it can be easily copied to discrete setting with our wDG framework, since there we might be demanded to use Hessians. Whether some difference of wDGs can work for approximating such Hessians or not is an interesting research topic, but we cannot say anything conclusive at this moment.
>
> >Question 3.  Can the limitations of the framework, such as the non-optimality of certain convergence rates, be overcome by considering alternative DEs and Lyapunov functionals?
>
> Thank you for pointing out this important point. Yes, we believe so. For example, in the example we showed in "Limitations",  the rate in our paper $O((1-\sqrt{\mu/L})^k)$ (NAG rate) comes from the continuous estimate in Thm. 2.5, the term $e^{-\sqrt{\mu}t}$. On the other hand, the optimal rate is known to be $O((1-\sqrt{\mu/L})^{2k})$ (a method was given in Scoy et al. (2017), which was then shown to be optimal in Drori--Taylor (2022); the arguments are more direct, i.e., not in the ODE approach).  In the ODE approach, if we find a combination of DE+Lyapunov functional that exhibits the rate $e^{-2\sqrt{\mu}t}$, then wDG shall be able to derive methods achieving the optimal rate. Several teams are now working to solve this open question.

---

> > ### Comment · Reviewer_7m3u · 2023-08-11
> > **Reviewer's response**
> >
> > Dear authors,
> >
> > Thank you for your clear response.
> >
> > While I still believe that the results in the paper were achievable because we already know the accelerated algorithm (especially how to define the wDG in the paper), the authors did convince me of the flexibility of their approach. As shown in the paper and the rebuttal, the paper potentially open new perspectives for novel analysis.
> >
> > I will raise my score to borderline accept - a strong accept would have been possible if there were some results that went beyond of what we already knew.
> >
> > Thank you again for your response and clarification!

---

> > > ### Author Response · Authors · 2023-08-17
> > >
> > > Thank you for your kind understanding.
> > > We understand your point that a new method better than existing methods would have made our paper more decisive. But at the moment, this is left as the most important open question in the whole ODE approach; we hope by our framework the search for better optimization ODEs will become more active research topic.

---

### Official Review · Reviewer_xD6M · 2023-06-27

**Soundness:** 3 good
**Presentation:** 3 good
**Contribution:** 2 fair
**Rating:** 6
**Confidence:** 3

**Summary:**

This paper focuses on the design of convex optimization schemes based on a general discretization framework applied to differential equations (DE). The authors heavily rely on Lyapunov based inequalities to provide convergence rates.
The authors build a systematic framework on top of the one proposed by Su, Boyd and Candès, allowing to provide an automatized analysis, thanks to the concept of weak discrete gradient (wDG). This weaker notion of DG allows one to overcome the assumptions needed for discrete versions of gradients in the DE approach. The resulting convergence rates turn out to be competitive, even though they might not be optimal.


**Strengths:**

The analysis of a large class of convex optimization algorithms can now be performed in an automatized fashion. Many existing optimization methods together with their rate estimates, can be retrieved as special cases of the authors' method. This offers a simpler perspective.

The presentation of the paper is great, the authors make an effort to avoid loosing the reader into too many technicalities. Not being an expert in such convex optimization schemes, it was still enjoyable to follow most of the details. The paper is well-written.


**Weaknesses:**

The estimated rates are not always optimal as demonstrated by the authors (some sub-cases of Theorem 5.5 for strongly convex functions), which indicates that there are still specific efforts to be pursued for certain algorithms.


**Questions:**

l81: The abbreviation PŁ is not defined in the main text. Please indicate what it stands for. (Polyak-Lojasiewicz)


**Limitations:**

There is a clear section dedicated to limitations with an explicit list:

- some methods do not fall into the authors' framework, e.g., Douglas–Rachford splitting method
- adjustment (for instance of time-stepping) could be improved for wDG
- the obtained rates are not always optimal

---

> ### Author Rebuttal · Authors · 2023-08-04
>
> Thank you very much for your careful reading and suggestions. We really appreciate it.
> Below are our responses to your criticisms in Weaknesses and Questions.
>
> >Weakness 1.  The estimated rates are not always optimal as demonstrated by the authors (some sub-cases of Theorem 5.5 for strongly convex functions), which indicates that there are still specific efforts to be pursued for certain algorithms.
>
> Thank you for pointing out this important point. We agree with your view that some rates in the present paper do not achieve known optimal rates, as we explicitly declared in Limitations. However, this comes from the limitation in the currently known "ODE + Lyapunov functional," not from our framework. Let us explain this by taking the example we showed in "Limitations." The rate in our paper $O((1-\sqrt{\mu/L})^k)$ (accelerate gradient method (AGM) rate) comes from the continuous estimate in Thm. 2.5, the term $e^{-\sqrt{\mu}t}$. On the other hand, the optimal rate is known to be $O((1-\sqrt{\mu/L})^{2k})$ (a method was given in Scoy et al. (2017), which was then shown to be optimal in Drori--Taylor (2022); the arguments are more direct, i.e., not in the ODE approach).  In the ODE approach, if we find a combination of DE+Lyapunov functional that exhibits the rate $e^{-2\sqrt{\mu}t}$, then wDG shall be able to derive methods achieving the optimal rate. To our best knowledge, such a combination is not yet known---but recently steady progress has been made. We the authors are aware of an ODE that seems promising, although we have not yet succeeded in finding a Lyapunov functional for it. Sun et al. (2020) derived a combination of ODE+Lyapunov functional that achieves a rate better than AGM, but it is still not optimal. In viewing this recent information, we expect to find the desired ODE+Lyapunov functional in the near future.
>
> >Question 1.  l81: The abbreviation PŁ is not defined in the main text. Please indicate what it stands for. (Polyak-Lojasiewicz)
>
> Thank you so much for your careful reading. We originally spelled out this, but we somehow dropped it in finalizing the submitted paper. We will appropriately include the full spell.

---

> > ### Comment · Reviewer_xD6M · 2023-08-17
> >
> > Thank you for your responses, I will retain my score.

---

### Official Review · Reviewer_tGA7 · 2023-07-01

**Soundness:** 3 good
**Presentation:** 4 excellent
**Contribution:** 3 good
**Rating:** 7
**Confidence:** 4

**Summary:**

The paper considers unconstrained convex optimization problems, studied from the angle of their close relation with differential equations. Specifically, the authors propose a framework for translating results from continuous time methods to their discrete time counterparts. They propose Discrete Gradients (a technical tool from Numerical Analysis) as a means for achieving this, and suitably adapt the concept to the convex optimization scenario (via weak Discrete Gradients). As such, the complexity of showing convergence for DE discretizations is mostly transferred to finding the appropriate wDG. The authors then show how this framework can recover known convex optimization results for some important methods in the class.

**Strengths:**

The paper follows in a line of work studying discrete gradient methods as optimization schemes. While the central concept of Discrete Gradients is not novel, its weak variant used in this paper and the unifying framework built around it are, to the best of my knowledge. The studied topic is of considerable interest to the optimization community, and the tools introduced in this work are welcome additions. The paper is written in a crisp and clear style (making for a very pleasant read), and the authors critically discuss their results in (mostly) adequate detail.

**Weaknesses:**

- Reference [1] seems to have quite some overlap in terms of results and considered classes of functions. In this light, a more detailed comparison is warranted in terms of e.g., assumptions -- especially on stepsize values, type of results, classes of functions, and techniques.
- The usage of the "free" iterate z (introduced in Def. 4.1) when devising the discrete accelerated schemes is opaque and would benefit from further discussion.

* Minor (typos and the like):
	- Line 315: of in
	- Line 315: the Hessian
	- you use (i), (ii) in Thm 4.2, but on page 4 line 131 you use the same notation for other concepts, which allows for confusion


[1] Ehrhardt, M. J., Riis, E. S., Ringholm, T., and Schönlieb, C.-B. (2018). A geometric integration approach to smooth optimisation: Foundations of the discrete gradient method. arXiv preprint, arXiv:1805.06444.

**Questions:**

- Can the authors provide some discussion on why, e.g., the max stepsize for Gradient Descent (line 1 in table 2) is more restrictive compared to the known strict upper bound of 2/L? It is not clear whether this kind of situation comes from some insurmountable limitations of the framework.
- Reference [2] is a missing reference for Lyapunov-based approaches. The paper has the same goal as the present work but achieves it via a different route, involving formulas for constructing Lyapunov functions. It would be good to briefly discuss it in the section "Relation to some other systematic/unified frameworks"
- Reference [3] might be relevant for motivating the unified approach

[2] De Sa, C. M., Kale, S., Lee, J. D., Sekhari, A., & Sridharan, K. (2022). From Gradient Flow on Population Loss to Learning with Stochastic Gradient Descent. Advances in Neural Information Processing Systems, 35, 30963-30976.
[3] Bansal, N., & Gupta, A. (2017). Potential-function proofs for first-order methods. arXiv preprint arXiv:1712.04581.

**Limitations:**

The authors address the limitations in a separate and very clearly written section and go into adequate detail describing them.

---

> ### Author Rebuttal · Authors · 2023-08-04
>
> Thank you very much for your careful reading and suggestions. We really appreciate it.
> Below are our responses to your criticisms in Weaknesses and Questions.
>
> >Weakness 1. Reference [1] seems to have quite some overlap in terms of results and considered classes of functions. In this light, a more detailed comparison is warranted in terms of e.g., assumptions -- especially on stepsize values, type of results, classes of functions, and techniques.
>
> Thank you so much for pointing out this important point. We would like to add the following paragraph in "Relation to some other systematic/unified frameworks":
>
> ```
> As said in Section 3, in the field of numerical analysis, the use of discrete gradients has been tried. Among them, Ehrhardt et al. (2018) is a pioneering work that comes with several theoretical results. Both this and the present work aim at convex, strongly-convex, and the PL functions (in the Appendix, in the present paper). The scope of Ehrhardt et al. (2018) was limited in the sense that they considered only discretizations of gradient flows with strict discrete gradients. Our target ODEs and discretizations are not limited to that, but as its price, our rate is worse in some strict discrete gradient discretizations of gradient flow. This comes from the difference in proof techniques: they proved convergence rates directly and algebraically, while our analysis is via Lyapunov functionals. They also gave several theoretical results besides the convergence analysis, such as the (unique) existence of solutions, and step-size analysis which are important in actual implementations. Whether these two framework could be unified would be an interesting future research topic.
> ```
>
> >Weakness 2. The usage of the "free" iterate z (introduced in Def. 4.1) when devising the discrete accelerated schemes is opaque and would benefit from further discussion.
>
> This is a very good point, and thank you for suggesting this.
> In fact, we can think of several possibilities for z, which are discussed in the proof of Thm. 5.4 (Appendix E.3) and Thm. 5.5 (E.4).
> Please note also that this can be regarded as a modification of the three points descent lemma (see, for example, https://fa.bianp.net/blog/2017/optimization-inequalities-cheatsheet/).
>
> We agree that our explanation about how we utilize z is not quite clear. We would like to add the following paragraph after Def. 4.1.
>
> ```
> Note that (8) can be regarded as a modification of the three points descent lemma, where the third variable z is utilized to give some estimates. The freedom in variable z in (8) is fully utilized also in this paper; see Thm. 5.4 and 5.5 and their proofs.
> ```
>
> >Weakness 3. Minor (typos and the like)
>
> Thank you so much for your very careful reading. We will fix them in the final version.
>
> >Question 1. Can the authors provide some discussion on why, e.g., the max stepsize for Gradient Descent (line 1 in table 2) is more restrictive compared to the known strict upper bound of 2/L? It is not clear whether this kind of situation comes from some insurmountable limitations of the framework.
>
> This is a very good point. We noticed this, and our understanding is as follows.
> As far as we know, the bound $2/L$ is shown in Nesterov (2004; the textbook, Thm 2.1.14), with the rate $c_1/(c_2 + k c_3)$ ($c_1, c_2, c_3$ are some constants), which is slightly different from the rate in our paper $c/k$ (although asymptotically they are the same).
> The rate $c/k$ is obtained in the ODE approach and the designated Lyapunov functional, but this time, we cannot have the strict bound $2/L$. In this sense, we understand this is a limitation of the ODE approach, or more specifically, the limitation in the known combination of ODE and Lyapunov functional. Whether there exists a better combination such that we can have the rate $c/k$ with the strict bound $2/L$ or not is an interesting open question.
>
> >Question 2.  Reference [2] is a missing reference for Lyapunov-based approaches. The paper has the same goal as the present work but achieves it via a different route, involving formulas for constructing Lyapunov functionals. It would be good to briefly discuss it in the section "Relation to some other systematic/unified frameworks"
>
> Thank you so much for letting us know of this. We would like to add the following paragraph in "Relation to some other systematic/unified frameworks"
>
> ```
> Another closely related work is Skhari et al. (2022), which proposed a framework to construct Lyapunov functionals for continuous ODEs. This is strong in view of the fact that generally Lyapunov functionals can be found only in ad hoc ways. Instead, they considered only the simplest gradient descent (and its stochastic version), while the main focus of the present paper lies in the discretizations.
> ```
>
> >Question 3.  Reference [3] might be relevant for motivating the unified approach
>
> Thank you for this information. We will include this reference in Introduction where we survey the history of the ODE and Lyapunov approach.

---

> > ### Comment · Reviewer_tGA7 · 2023-08-10
> > **Thank you for your response**
> >
> > Thank you for your response to the questions. I will maintain my score.

---

> > > ### Author Response · Authors · 2023-08-17
> > >
> > > Thank you so much for your quickest and warm response.  We sincerely thank all of your kind effort.

---

### Official Review · Reviewer_eJSN · 2023-07-07

**Soundness:** 4 excellent
**Presentation:** 3 good
**Contribution:** 3 good
**Rating:** 6
**Confidence:** 3

**Summary:**

This paper introduces a family of oracles called wDG (weak discrete gradient) verifying (8).
This family constraint (8) has been created in order to make proof works in the discrete setting and has been inspired by the observation of what happens in the continuous setting.
As expected, authors propose results obtained running classical algorithms replacing the gradient by any of those oracles. This new framework allows including many popular algorithms using different oracles such as proximal operator.

**Strengths:**

The paper is concise and clear. The problem is motivated, and well explained.
The new framework is fairly large and authors detailed examples of classical algorithms that are included in their framework.

**Weaknesses:**

- It might be because I did not know this line of work, but based on the title, I thought of a completely different result, on how, using gradients on the current iterate, efficiently discretize an ODE. Finally, the proposed framework seems to be more about the nature of the oracle itself, not the algorithm, and some of them (prox, average gradients, ...) are not always accessible.

- Thm 2.3: From this thm, it seems that the convergence rate can be arbitrarily good. Authors should discuss this here and mention that discretization error unables to converge faster than the proven lower bound $1/t^2$.

$\underline{\text{Typos:}}$

- l.23: « strong convex » -> strongly
- l.164: « in of »
- eq 10: I think \beta and \gamma have been permuted
- Thm 5.2: « Let $\bar{\nabla} f$ be a wDG of $f$ […] let f be […] » -> « Let f be … let $\bar{\nabla} f$ be …. »

**Questions:**

NA

---

> ### Author Rebuttal · Authors · 2023-08-04
>
> Thank you very much for your careful reading and suggestions. We really appreciate it.
> Below are our responses to your criticisms in Weaknesses.
>
> >Weakness 1. It might be because I did not know this line of work, but based on the title, I thought of a completely different result, on how, using gradients on the current iterate, efficiently discretize an ODE. Finally, the proposed framework seems to be more about the nature of the oracle itself, not the algorithm, and some of them (prox, average gradients, ...) are not always accessible.
>
> A. We agree with the comment that if we regard weak discrete gradient as an oracle, the main focus in our paper is exactly "the nature of the oracle itself," i.e., we investigate conditions wDGs should satisfy.
> We could not catch what the reviewer meant by the second sentence. Below are some possibilities we considered.
>
> First, if the reviewer means that some wDGs (prox, average gradients, ...) are not acceptable as oracles for first-order methods because they lead to implicit methods, we would like to claim that this does not immediately mean that these wDGs are impractical. For example, for minimization problems with a regularization term, one can generate variants of the accelerated proximal gradient method (with the convergence rate guaranteed) by considering wDGs for the regularization term. We have experimentally confirmed that some of these methods perform better than the accelerated gradient method for some objective functions.
>
> Second, if the problem is in that some wDGs seem to die for some objective functions (in our paper, (v) and (vi) in Thm 4.2 seem to work only for strongly-convex functions), we hope to claim that this does not immediately mean the limitation of our framework, but rather clarifies that some choices of wDG are only of limited use, which is informative for users of our framework.
>
> If both understandings are not what the reviewer intended, it would be greatly appreciated if the reviewer could kindly rephrase the criticism. Then we are more than happy to add further response.
>
> >Weakness 2. Thm 2.3: From this thm, it seems that the convergence rate can be arbitrarily good. Authors should discuss this here and mention that discretization error unables to converge faster than the proven lower bound 1/t^2.
>
> A. Thank you for pointing out this very important point. Yes, as the reviewer says, even though in the continuous ODE arbitrarily high convergence rate seems possible, it does not at all mean that we can draw discrete schemes (actual optimization methods) out of the ODE while keeping the same high order. What happens actually is that, greedily demanding a higher rate is penalized at the timing of discretization from the numerical stability. See, for example,
>
> Ushiyama, K., Sato, S., and Matsuo, T., Essential convergence rate of ordinary differential equations appearing in optimization, JSIAM Letters, 14 (2022). https://doi.org/10.14495/jsiaml.14.119
>
> We would like to add a new remark after Thm 2.3 as follows.
>
> ```
> Rem 2.X. From Thm 2.3 it might seem that we can achieve arbitrarily high order rate. Although it is surely true in the continuous context, it does not imply we can construct discrete optimization methods from the ODE. In fact, greedily demanding a higher rate is penalized at the timing of discretization from the numerical stability. See, for example, the discussion in Ushiyama--Sato--Matsuo (2022).
> ```
>
> Thank you also for pointing out the typos. We would like to thank you again for your very careful reading.

---

> > ### Comment · Reviewer_eJSN · 2023-08-19
> > **Thank you**
> >
> > I thank the authors for their answers. I only had a few minor concerns and authors acknowledged they will fix those in the camera ready version. I keep thinking this paper is nice and I would like to see it at NeurIPS.

---

### Decision · Program_Chairs · 2023-09-21

**Decision:**

Accept (poster)

**Comment:**

All reviewers are positive about the paper. Please revise the paper and add a discussion about the practicability of your method.